# MAPK-pathway inhibition mediates inflammatory reprogramming and sensitizes tumors to targeted activation of innate immunity sensor RIG-I

Johannes Brägelmann [1,2,3,4,24 ✉], Carina Lorenz[1,2,3,24], Sven Borchmann[3,5,6], Kazuya Nishii[7], Julia Wegner [8], Lydia Meder[3,4,5], Jenny Ostendorp[1,2,3], David F. Ast[1,2,3,4], Alena Heimsoeth[1,2,3], Takamasa Nakasuka[7], Atsuko Hirabae[7], Sachi Okawa[7], Marcel A. Dammert [1,2,3], Dennis Plenker [9,10], Sebastian Klein[6,11], Philipp Lohneis [11], Jianing Gu[12,13], Laura K. Godfrey[12,13], Jan Forster[13,14], Marija Trajkovic-Arsic[12,13], Thomas Zillinger [8], Mareike Haarmann[4], Alexander Quaas[11], Stefanie Lennartz[1,2,3], Marcel Schmiel[2,11], Joshua D'Rozario [2], Emily S. Thomas[2,15,16], Henry Li [17], Clemens A. Schmitt [18,19,20], Julie George [2,21], Roman K. Thomas[2,11,22], Silvia von Karstedt [2,15], Gunther Hartmann [8], Reinhard Büttner [11], Roland T. Ullrich[3,5], Jens T. Siveke [12,13], Kadoaki Ohashi[7,23], Martin Schlee [8] & Martin L. Sos [1,2,3 ✉]

Kinase inhibitors suppress the growth of oncogene driven cancer but also enforce the selection of treatment resistant cells that are thought to promote tumor relapse in patients. Here, we report transcriptomic and functional genomics analyses of cells and tumors within their microenvironment across different genotypes that persist during kinase inhibitor treatment. We uncover a conserved, MAPK/IRF1-mediated inflammatory response in tumors that undergo stemness- and senescence-associated reprogramming. In these tumor cells, activation of the innate immunity sensor RIG-I via its agonist IVT4, triggers an interferon and a pro-apoptotic response that synergize with concomitant kinase inhibition. In humanized lung cancer xenografts and a syngeneic *Egfr*-driven lung cancer model these effects translate into reduction of exhausted CD8$^+$ T cells and robust tumor shrinkage. Overall, the mechanistic understanding of MAPK/IRF1-mediated intratumoral reprogramming may ultimately prolong the efficacy of targeted drugs in genetically defined cancer patients.

A full list of author affiliations appears at the end of the paper.

For cancer patients, the advent of precision medicine aiming at oncogenic mutations in kinases like EGFR and BRAF led to a significant improvement of the overall survival across different tumor types[1–3]. However, in virtually all patients, drug response is followed by therapy resistance and disease progression. Tumor recurrence is thought to result from cancer cells that evade cell death and persist during the treatment, a process that has also been observed in preclinical models of various cancer types. In these models drug, persistent cells remain viable in a quiescent state of limited proliferative capacity and several mechanisms have been proposed how cells may enter this persistent state[4–8].

Increasing evidence indicates that therapeutic stresses drive tumor cells into senescence, beneficial as an arrest, but potentially detrimental due to other senescence-associated (SA) effects[4,6,9,10]. This process may involve the reprogramming of tumor cells and activation of innate immunity networks that may subsequently rewire the interaction of tumor cells and immune cells within their microenvironment[11,12]. It has been shown that compounds such as cyclin-dependent kinase (CDK) inhibitors and epigenetic drugs may have the capacity to evoke similar effects through the induction of inflammation in the tumor-microenvironment[11,13–17]. These effects were in part explained by the engagement of nucleic acid receptors (NARs) and their downstream STING- or MAVS-dependent innate immunity pathways[13–15,18,19]. NARs like cGAS, MDA5, TLR3, or RIG-I (DDX58) are conserved proteins of the cellular innate immune system aiming to recognize foreign nucleic acids e.g. following a virus infection to induce cell death or improved immune recognition[20,21]. Exploiting their role in stimulating immune recognition, NAR agonists have been used successfully to promote response to immunotherapies[22,23]. In addition, stimulation of NARs like RIG-I with synthetic agonists has been used to directly target cancer cells and has been shown to induce tumor cell death in pre-clinical models, which offers additional treatment options[24–26].

Here, we uncover a MAPK/IRF1-mediated transcriptional reprogramming of senescent cancer cells that survive drug treatment. We further define the role of inflammatory signaling in drug-tolerant cells including upregulation of NARs and the associated immune infiltration in vivo. Finally, we show that the RIG-I agonist IVT4 specifically exploits the reprogrammed drug-tolerant state and limits the outgrowth of oncogene-dependent tumor types in vitro and invivo.

## Results

**Targeted drugs trigger an inflammatory program in cell-cycle arrested cells.** To study the adaptive changes during targeted inhibition of oncogenic signaling we exposed cancer cells ($n = 9$) driven by diverse activated kinases (EGFR, MET, BRAF, HER2, ALK) to targeted drugs and performed transcriptional profiling using RNA-seq (Supplementary Fig. 1a). Despite considerable transcriptional heterogeneity at baseline, a combined analysis revealed strong repression of genes related to MYC signaling and cell cycle progression (Fig. 1a, Supplementary Fig. S1b, Supplementary Table 1). Remarkably, we also observed a shared induction of Interferon (IFN) target gene sets[13,14] and an adult tissue stem-cell gene signature (ATSC)[27], which has recently been associated with therapy-induced senescence in the joint analysis (Fig. 1a)[28]. BRAF-mutant and EGFR-mutant cells showed a tendency towards more pronounced transcriptional effects compared to cell lines driven by other oncogenes (Supplementary Fig. 1b). As expected, kinase inhibitor treatment led to caspase-dependent cell death in a subset of cells (Supplementary Fig. 1c, d) while a large fraction of cells arrested in a G1/G0 cell-cycle state and remained viable for a follow-up time of at least 15 days (Fig. 1b,

Supplementary Fig. 1e)[4,29]. Of note, in osimertinib-treated EGFR-mutant cells, we identified induction of p27 as a potential mediator of cell cycle arrest (Fig. 1b, Supplementary Fig. 1f)[28]. Across different cell lines and treatments, drug-tolerant cells exhibited characteristics of therapy-induced senescence such as positive β-galactosidase staining, a delayed secretion of Interleukin-6 (IL-6), higher levels of BH3 family members and histone marks H3K9me3 and H3K27me3 (Supplementary Fig. 1h-m)[10,30]. To assess the time dynamics of the observed changes we performed time-series expression profiling of three independent EGFR-mutant cell lines and revealed a rapid induction of ATSC and IFN-signaling and suppression of MYC- and E2F-targets (Fig. 1c, d, Supplementary Fig. 1n). To investigate the observed changes on a single-cell level we performed single-cell RNA-sequencing (scRNA-seq) of osimertinib-treated PC9 and vemurafenib-treated Colo205 cells. Clustering based on most variable genes and based on ATSC genes however clearly separated treated and untreated cells (Supplementary Fig. 2a, b). The scRNA-seq confirmed downregulation of relevant senescence-repressed markers[30] such as HMGB1 and HMGB2 and induction of a subset of known senescence-associated secretory phenotype (SASP) genes (Supplementary Fig. 2c). Furthermore, pseudo-time analysis[31] of untreated PC9 cells based on genes expressed in G1/S or G2/M[32] did not show a consistent pattern of expression of SA genes in various cell cycle phases. These findings are compatible with a model in which cells undergo a treatment-dependent reprogramming and are in line with previous results obtained in similar settings (Supplementary Fig. 2d, e)[28,29]. To investigate whether similar changes appear in vivo we extended our analyses to two independent EGFR-mutant patient-derived xenografts (PDX), which displayed a similar upregulation of inflammatory signaling and ATSC during the response to EGFR inhibition (Fig. 1e, Supplementary Fig. 2f).

We next asked whether the induction of an inflammatory and SA stemness response including SASP components in drug-tolerant cells may also have an effect on the tumor microenvironment. We therefore employed a humanized PC9 xenograft model and observed an increase of infiltrating T-cells following osimertinib treatment (Fig. 1f, g, Supplementary Fig. 2g). Using the Hyperion imaging system we validated higher lymphocyte infiltration with osimertinib treatment (Supplementary Fig. 3a–c) and also observed a shift of proliferative activity (Ki67) from tumor cells to CD3+ cells and cytolytic activity as measured by co-staining of CD8 and Granzyme B under treatment (Fig. 1g, Supplementary Fig. 3a–c). Similarly, CD8 cells were significantly less exhausted as measured by a lower rate of TIM3+ and PD1+CD69− CD8 cells after osimertinib treatment in flow cytometry (Fig. 1i). We subsequently validated our findings of transcriptional adaption mechanisms by comparing public gene expression array data of HER2-amplified breast cancer patients undergoing neoadjuvant treatment with the monoclonal antibody trastuzumab[33]. In this cohort, we identified a very similar induction of ATSC and IFN programs when comparing responders vs. non-responders following neoadjuvant therapy and comparable trends in baseline to post-treatment analysis in responders and non-responders (Supplementary Fig. 3d). In addition, we analyzed public RNA-seq data of BRAF-mutant melanoma patients containing samples before BRAF or BRAF+MEK inhibitor treatment, during therapy, and after resistance had developed[34]. In line with our cell line and PDX data, we observed a significant induction of IFN, ATSC, and repression of MYC and E2F signatures when comparing pre- vs. on-treatment samples (Fig. 1j). Furthermore, we found an enrichment of samples on treatment for a senescence-like expression profile[30] similar to the phenotype observed in Colo205 and PC9 cells (Fig. S3e). To assess the infiltration with tumor-associated lymphocytes we inferred the immune cell

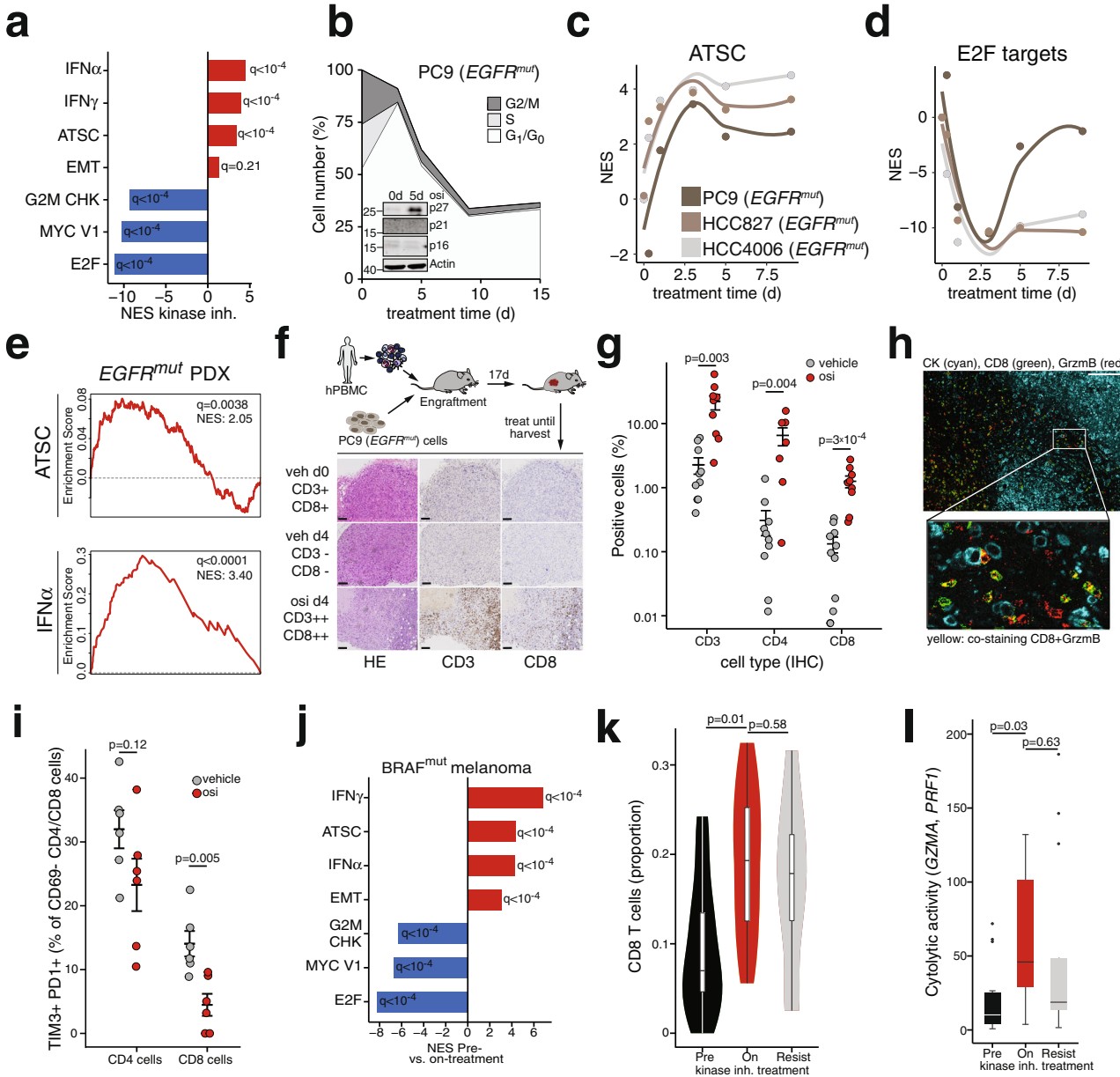

**Fig. 1 Kinase inhibition induces senescence-associated inflammatory signaling. a** Core gene set of a combined gene set enrichment analysis (GSEA) in oncogene-driven cancer cell lines (PC9, HCC827, HCC4006, H1993, H3122, A375, Colo205) after 3 days of treatment with their respective kinase inhibitor. **b** Cell count of *EGFR^mut* PC9 cells under osimertinib (osi, 300 nM). The upper line indicates normalized cell number, shaded areas corresponding cell cycle distribution (n = 3). Inset: Immunoblot of cell cycle regulator genes in PC9 cells after 5 days of osimertinib treatment. **c, d** GSEA of time-series RNAseq in *EGFR^mut* cells indicates temporal adaptation processes (ATSC = adult tissue stem cell gene set, NES = Normalized enrichment score). **e** GSEA of the ATSC and the IFNα gene set across the two *EGFR^mut* PDX models (osi vs. vehicle). **f** Schematic of the humanized mouse model (top) and exemplary histology of low (−), medium (+), and high (++) CD8 T cell infiltration (bottom). Scale bar 100 µm, representative images of in total n = 19 tumors of n = 10 mice. **g** Digital pathology-based quantification of T cell infiltration in humanized mice following 4 days of treatment with osimertinib (5 mg/kg, n = 9 tumors) or vehicle (n = 10 tumors) (error bars indicate mean ± SEM). **h** Hyperion imaging mass cytometry false-color image of an osimertinib treated tumor from (**f**) stained for pan-cytokeratin (CK, cyan), CD8 (green), and Granzyme B (red). The overlay of CD8 and red is colored yellow. Scale bar 100 µm, representative image of n = 6 regions. **i** Flow cytometry analysis of infiltrating T cells in humanized PC9 xenografts after 4 days of treatment (in total n = 6 tumors of n = 3 mice per group; error bars indicate mean ± SEM). **j** RNA-seq-based GSEA of public *BRAF^mut* melanoma patient data, comparing patients before (Pre-treatment) with patients during BRAF or BRAF + MEK inhibition (On-treatment). **k** Proportion of CD8 T cell infiltration inferred from bulk RNA-seq in the melanoma patients from (**j**) sequenced before (Pre, n = 11) or during (On, n = 11) kinase inhibition or after resistance (Resist, n = 10) had developed. **l** Cytolytic activity as the geometric mean of granzyme A and perforin RNA expression in patients from (**j**). Significance was calculated by t tests in (**g**), (**i**), (**k**) (**l**) and Kolmogorov–Smirnov-based permutation test as FDR-corrected q-values in (**a**), (**e**), (**j**). All tests are two-sided. Boxplots display median (center line), 25th/75th percentile (lower/upper box hinges), whiskers extend to the most extreme value within 1.5× interquartile range (IQR) of the hinges. Source data are provided as a Source Data file.

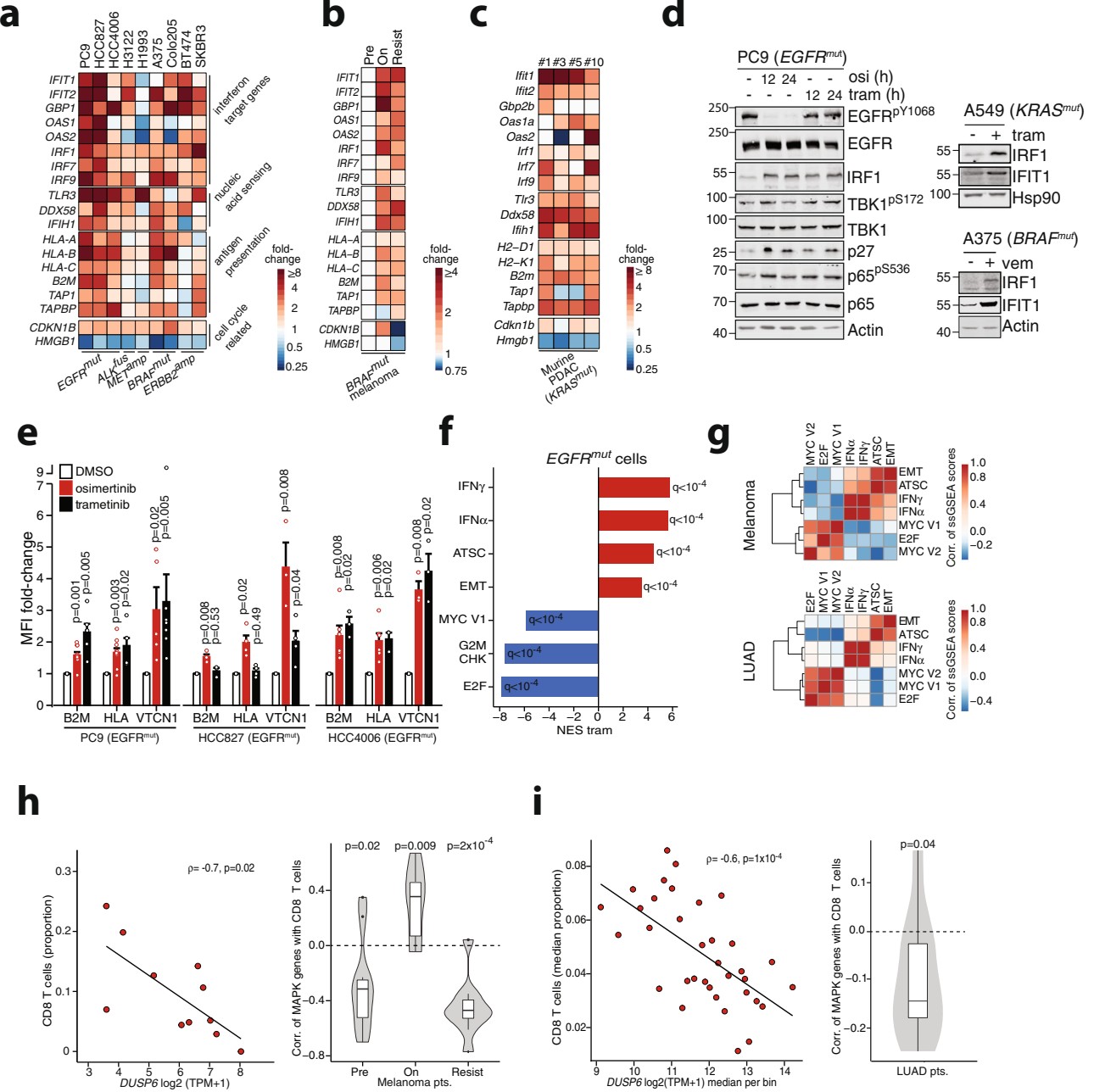

composition from RNA-seq using CIBERSORT[35]. Comparable to the humanized mouse model we observed a significantly higher proportion of CD8 T-cells and NK-cells in on-treatment compared to pre-treatment samples (Fig. 1k, Supplementary Fig. 3f, g). Accordingly, cytolytic activity as measured by an RNA-seq-based score[36] was significantly higher in on-treatment samples (Fig. 1l). Interestingly, the higher influx of CD8 cells in matched pre-treatment vs. on-treatment biopsies in these patients significantly correlated with a longer time to progression (Supplementary Fig. 3h). This further supports a model in which the treatment-induced reprogramming of drug-tolerant cells promotes inflammatory signaling including an associated response of the tumor microenvironment.

**MAPK inhibition is sufficient to induce inflammatory programs in cell-cycle arrested cells.** We next investigated the specific genes that may be associated with the observed drug-induced phenotype. Across our cell line panel, we observed a broad

consensus in the upregulation of genes involved in IFN signaling, nucleic acid-sensing, and antigen presentation (Fig. 2a). Similar inductions of target genes could also be validated in the melanoma patients and in a previously published dataset of *RET*-rearranged LC-2/AD cells during RET inhibition[37] (Figs. 2b and S4a). Moreover, cells derived from a KRAS-driven pancreatic ductal adenocarcinoma (PDAC) genetically engineered mouse model (GEMM)[38] revealed an inflammatory reprogramming upon trametinib treatment (Fig. 2c, Supplementary Fig. 4b). Surprisingly, despite the rapid upregulation of IFN-target genes, no relevant secretion of IFN was detectable in the supernatant of *EGFR*-mutant cells treated with osimertinib (Supplementary Fig. 4c, d). Also, a knock-out of the IFN receptor (*IFNAR1*) did not affect the osimertinib-induced activation of IFIT1 or IRF1, while it largely abrogated the response to IFNα in PC9 cells (Supplementary Fig. 4e). Interestingly, IFNα did not induce IRF1 expression further supporting the notion that kinase inhibitor effects are IFN-independent. It has been proposed previously that

**Fig. 2 MAPK-pathway mediates inflammatory signaling and immune escape. a** Fold-changes by RNAseq expression analysis following 3 days kinase inhibitor treatment in oncogene-driven cancer cell lines. **b** Fold-changes in *BRAF*[mut] melanoma patients sequenced before (Pre) or during (On) kinase inhibition or after resistance (Resist) to BRAF or BRAF + MEK inhibition. **c** Fold-changes in primary cells derived from a *KRAS*[mut] PDAC GEMM after 48 h treatment with trametinib compared to controls. **d** Left: Immunoblot of key inflammatory signaling nodes in *EGFR*[mut] PC9 cells treated for 12 or 24 h with osimertinib (osi, 300 nM) or trametinib (tram, 100 nM). Right: Immunoblot of treated *KRAS*[mut] A549 (trametinib, 100 nM) and *BRAF*[mut] A375 (vemurafenib, 1 μM) for 3 days. Each representative blot of $n = 3$ independent experiments. **e** FACS estimation of surface expression in *EGFR*[mut] cells following 3 days treatment with osimertinib (300 nM) or trametinib (100 nM). mean fluorescent intensity (MFI) as fold-change normalized to DMSO controls. Bars display mean ± SEM of independent biological replicates (PC9 B2M/HLA: osi $n = 10$, tram $n = 5$; VTCN1: osi $n = 5$, tram $n = 8$; HCC827 B2M/HLA: osi/tram $n = 4$, VTCN1 osi $n = 3$, tram $n = 4$; HCC4006 HLA/B2M osi $n = 6$, tram $n = 3$, VTCN1 osi/tram $n = 3$); *P*-values adjusted by Benjamini–Hochberg. **f** Combined GSEA of RNA-seq from PC9, HCC827 and HCC4006 cells treated with trametinib (100 nM, 72 h). (Significance as FDR-corrected *q*-values). **g** Pairwise correlations of single-sample (ss) GSEA scores for key gene sets in RNA-seq of untreated BRAF[mut] melanoma patients ($n = 14$, top) or TCGA lung adenocarcinoma patients (LUAD, $n = 515$, bottom). Color indicates Pearson correlation coefficients. **h** Left: Correlation of RNA-seq inferred CD8 T cell infiltration with an expression of the negative MAPK feedback regulator *DUSP6* in untreated *BRAF*[mut] melanoma patients ($n = 11$) (TPM = transcripts per million). Right: Distribution of individual correlations of CD8 T cell proportion with an extended set of MAPK activity genes[45] in patients from (**b** and Fig. 1k). Distribution of the $n = 10$ genes' correlation coefficients with CD8 T cell proportion was tested for significance using one-sample *t* tests adjusted with Bonferroni–Holm. **i** Left: Correlation of RNA-seq-based CD8 T cell infiltration with DUSP6 in untreated TCGA lung adenocarcinoma patients ($n = 350$) grouped as $n = 10$ patients per bin and normalized expression/CD8 T cell proportion as median per bin. Right: Correlation of CD8 T cell proportion with genes of the extended $n = 10$ MAPK genes in unbinned patients ($n = 350$). Significance was calculated with two-sided paired *t* tests for log fold-changes (**e**) and one-sample *t* tests in (**h**, right) and (**i**, right). Spearman correlation was used in (**h**, **i**). Boxplots display median (center line), 25th/75th percentile (lower/upper box hinges), whiskers extend to the most extreme value within 1.5× interquartile range (IQR) of the hinges. Data points beyond the whiskers are displayed individually. Source data are provided as a Source Data file.

RAS/MEK signaling suppresses IFN-induced transcription via downregulation of IRF1 and we speculated that targeted inhibition of oncogenic signaling could reverse this phenotype[39]. Indeed, targeted MAPK inhibition with trametinib largely recapitulated the rapid kinase inhibitor-mediated induction of IRF1 and IFIT1 (Fig. 2d, Supplementary Fig. 4f–h). In contrast, these effects were not detectable (A375, H3122) or less pronounced (PC9, HCC4006, Colo205) when using the PI3K inhibitor apitolisib (Supplementary Fig. 4g,h). Similarly, MEK inhibition in *KRAS*-mutant lung cancer A549 cells and BRAF inhibition in *BRAF*-mutant melanoma A375 cells also increased IRF1 and IFIT1 protein levels (Fig. 2d, right). A similar trend was evident in *KRAS*[G12C]-mutant (NRAS-KO) Lewis lung carcinoma cell lines[40] when treated with a KRAS-G12C inhibitor AMG510 or trametinib (Fig. S4i). Accordingly, inhibition downstream of MEK using the ERK-inhibitor SCH779284 induced IRF1 and IFIT1 expression in PC9 cells (Supplementary Fig. 4j). Among the genes induced by kinase inhibitor treatment were also several members of the antigen-presenting machinery, which may diminish the ability of tumor cells to evade the immune system (Fig. 2a). However, we also noticed upregulation of several proteins previously linked with resistance to immune checkpoint blockade (ICB) such as VTCN1 (B7-H4) across human cell lines and murine PDAC cells (Supplementary Fig. 4k,l)[41,42]. Indeed, we validated the increased surface expression of HLA, B2M, and VTCN1 on *EGFR*[mut] cells which was in accordance with corresponding RNA levels after osimertinib and similar responses after trametinib treatment (Fig. 2e, Supplementary Fig. 4m). In contrast, PD-L1 was decreased with treatment in vitro and in vivo (Supplementary Fig. 4n,o). Also, more generally, the gene sets modulated upon trametinib treatment in *EGFR*-mutant cells largely resembled the reprogramming induced by EGFR inhibition with a strong increase if IFN target genes (Fig. 2f). An increase of VTCN1 protein expression was moreover evident in several additional cell lines ($n = 5$ of 6) when treated with their respective kinase inhibitors (Supplementary Fig. 4p). This indicates that inhibition of the driving oncogene or of MAPK signaling can alter tumor-innate inflammatory pathways that may impact tumor-immune interactions at various levels.

Considering the drug-induced antagonistic association between gene sets related to IFN signaling and stemness on the one hand and cell cycle arrest markers on the other (E2F, MYC, G2M

CHK) (Figs. 1a, i and S3d) we speculated that this may reflect a general inverse association that may be also present in untreated primary patient samples. Indeed, correlation of single-sample gene set enrichment analysis (ssGSEA) results in lung adenocarcinoma tumor (LUAD) samples and treatment-naïve *BRAF*-mutant melanoma patients revealed that patients with a strong enrichment of cell cycle-related gene sets had lower enrichment of inflammatory and stemness signature and vice versa (Fig. 2g). Furthermore, we observed a negative correlation of DUSP6 and SPRY4, two negative feedback regulators of MAPK signaling[43,44] with the presence of CD8[+] T-cells in pre-, but not in on-treatment *BRAF*-mutant melanoma patients (Fig. 2h, left, Supplementary Fig. 4q–s). Negative correlations with CD8[+] T-cells were also observed when extending the analysis to a gene panel of a recently proposed MAPK pathway activation score containing genes upregulated with MAPK activity[45]. These results showed also that high MAPK activation was associated with decreased presence of CD8[+] T-cells (Fig. 2h, right). We obtained similar results when we analyzed RNA-seq data of untreated LUAD patients (Fig. 2i, Supplementary Fig. 4t). Thus, in oncogene-dependent tumors, MAPK activity may promote an immune evasive, proliferative state that can be reversed through targeted inhibition of oncogene signaling.

**Drug-induced IRF1 expression promotes activation of intratumoral immune signaling.** In the next step, we aimed to dissect the role of MAPK signaling for the induction of IFN target genes and potential cross-talk with cell death programs. In osimertinib-treated *EGFR*-mutant cells we did not observe a difference in expression of key IFN target genes with and without caspase inhibition (Fig. 3a, Supplementary Fig. 5a). Since caspase-inhibition prevents osimertinib-induced cell death (Supplementary Fig. 1d) this indicates that cell death is dispensable in the process. Similarly, treatment with the chemotherapeutic taxol (paclitaxel) reduced cell viability ($GI_{50} = 16$ nM), but did not induce *IRF1* or *IFIT1* in PC9 cells (Fig. 3b, Supplementary Fig. 5b). This data indicates that inhibition of the MAPK pathway and not cell death leads to inflammatory reprogramming. To further characterize these effects we genome-edited PC9 cells using CRISPR/Cas9 to generate osimertinib-resistant PC9[T790M+C797S] cells (Fig. 3c, Supplementary Fig. 5c, d)[46]. As expected, robust induction of IFN and stemness genes in RNA-Seq of

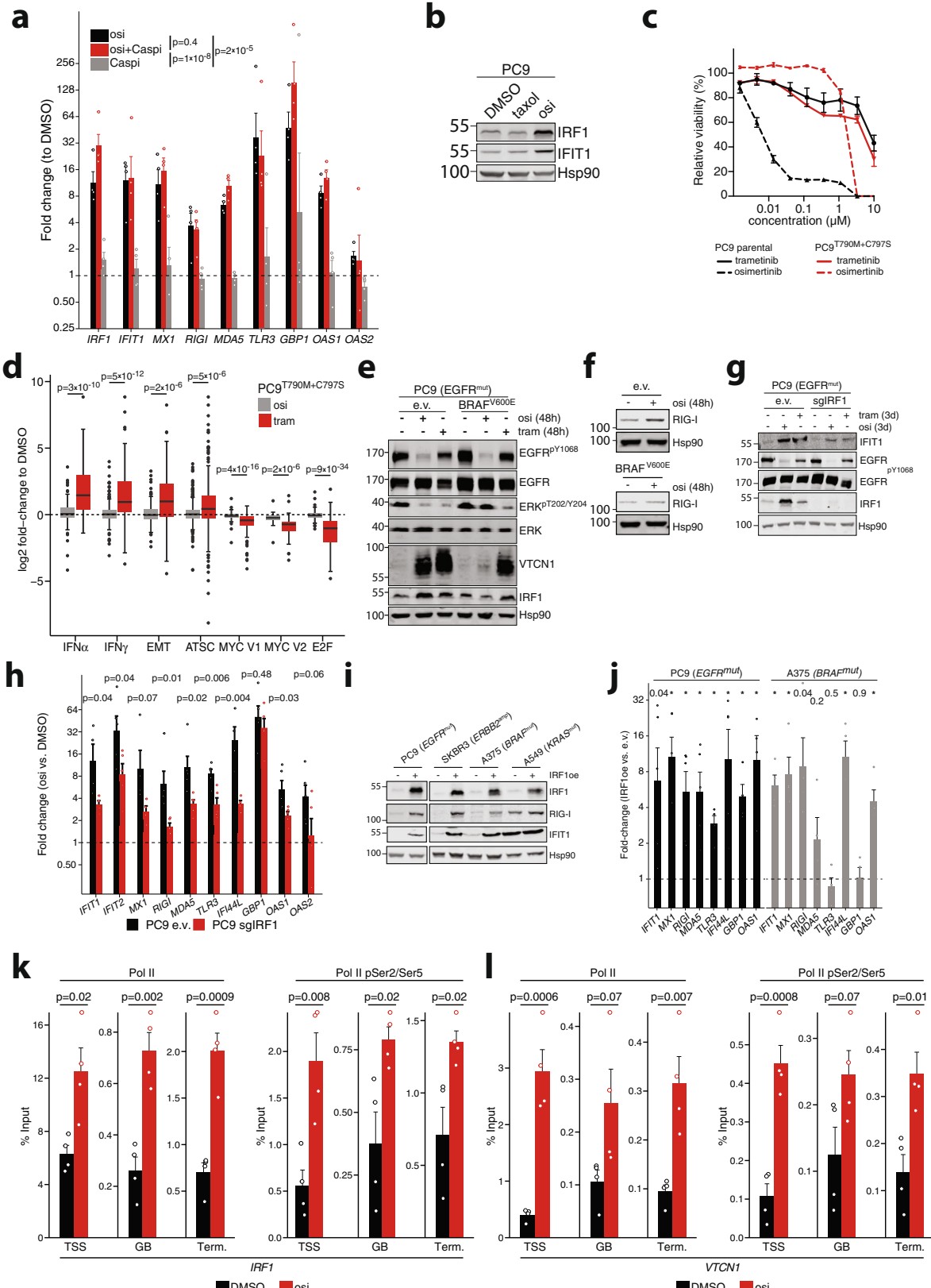

PC9$^{T790M+C797S}$ cells was no longer achievable by osimertinib but with the MEK inhibitor trametinib (Fig. 3d, Supplementary Fig. 5e).

To specifically prevent MAPK inhibition during osimertinib treatment, we generated PC9 cells that stably overexpress *BRAF$^{V600E}$*. Here, osimertinib treatment did not lead to induction of *IRF1, VTCN1,* and of the nucleic-acid sensor RIG-I (*DDX58*)

(Fig. 3e, f). The re-analysis of a published gene expression dataset of non-malignant human bronchial epithelial cells (HBEC) further supports a model in which MAPK signaling controls inflammation, as the induction of *KRAS* G12D expression led to a suppression of the IFNα gene set (Supplementary Fig. 5f)[47]. We next performed CRISPR/Cas9-based *IRF1*-knockouts in PC9

**Fig. 3 Inflammatory transcription is driven by the MAPK–IRF1 axis. a** RT-qPCR analysis of IFN-target genes of PC9 cells treated with osi (300 nM), Caspase inhibitor (Caspi; Q-VD-OPH, 10 μM) or a combination of both for 72 h. Fold-change compared to DMSO controls (mean ± SEM of $n = 4$ independent biological replicates). **b** Immunoblot of PC9 cells treated with osi (300 nM) or taxol (30 nM) for 48 h. Representative image of $n = 3$ independent experiments. **c** Viability of treated parental PC9 and CRISPR-edited PC9$^{T790M+C797S}$ (72 h, mean ± SEM of $n = 3$ independent biological replicates). **d** RNA-seq-based expression changes in relevant gene sets in treated vs. control PC9$^{T790M+C797S}$ (72 h with $n = 2$ replicates across $n = 2$ conditions). Boxplots display median (center line), 25th/75th percentile (lower/upper box hinges), whiskers extend to the most extreme value within 1.5× interquartile range (IQR) of the hinges. Data points beyond the whiskers are displayed individually. **e, f** Immunoblots of PC9-e.v. control or PC9-$BRAF^{V600E}$ cells following 48 h treatment with osimertinib or trametinib. Representative images of $n = 3$ independent experiments each. **g** Immunoblot of PC9 carrying lentiCRISPRv2 empty vector (e.v.) or an IRF1-KO (sgIRF1). **h** RT-qPCR analysis in osimertinib treated (300 nM, 72 h) cells from (**g**). Fold-changes were calculated as 2^ddCt compared to DMSO control and normalized to GAPDH. Mean ± SEM of independent biological replicates ($n = 4$ MX1, IFIT1, IFI44L, $n = 5$ other genes). **i** Immunoblot after transfection with IRF1 overexpression (oe) or empty vector (e.v.) plasmid for 72 h. Representative image of $n = 4$ independent experiments. **j** RT-qPCR analyses after 72 h IRF1 overexpression in PC9 and A375 ($n = 5$ and $n = 4$ independent transfection experiments, respectively). Fold-changes are calculated as 2^ddCt compared to e.v. control and normalized to GAPDH, Benjamini–Hochberg corrected p-values shown, *$p = 0.02$; Bars indicate mean ± SEM. **k, l** ChIP-qPCR analysis of total Pol II or phosphorylated Pol II pSer2/5 RNA Pol II binding to the transcription start site (TSS), gene body (GB) and transcription termination site (Term.) of IRF1 and VTCN compared to input control after 72 h osimertinib (300 nM) or DMSO treatment ($n = 4$ independent biological replicates, mean ± SEM). Significance of global treatment differences calculated on log-fold changes using Tukey-post hoc test after two-way ANOVA adjusting for gene-specific effects in (**a**). Significance calculated by two-sided t tests on log-fold changes with Benjamini–Hochberg adjustment for multiple testing in (**d, j**) and by two-sided t tests in (**h, k, l**). Source data are provided as a Source Data file.

cells, which suppressed trametinib-induced or osimertinib-induced upregulation of not only IRF1 but also of IFIT1 (Fig. 3g). Of note, the rapid and strong de-repression of *IRF1* (PC9, HCC827, HCC4006 cells) was partially accompanied by other IRF family members, which may potentially also contribute to the total IFN target gene output (Fig. 2a, Supplementary Fig. 5g). Nevertheless, *IRF1* knock-out significantly reduced transcriptional induction of several IFN target genes compared to e.v. control cells when treated with osimertinib, but not other IRFs (Fig. 3h, Supplementary Fig. 5h). In a complementary approach, transient overexpression of IRF1-induced IFN target genes in several cell lines indicating that IRF1 alone may suffice in specific contexts (Fig. 3i, j). Furthermore, using ChIP analyses we identified significant recruitment of RNA polymerase II (Pol II) and Pol II p-Ser2/Ser5 to the transcription start site (TSS), the gene body (GB) and at the transcription termination of *IRF1* and *VTCN1* upon 72 h osimertinib treatment (Fig. 3k, l). This Pol II pattern is indicative of active transcription in contrast to the osimertinib-induced repression of transcription of the negative MAPK-feedback regulator *DUSP6* (Supplementary Fig. 5i). Taken together, in oncogene-driven cells inhibition of MAPK pathway may be able to de-repress *IRF1* transcription that mediates activation of downstream inflammatory programs.

**Activation of RIG-I triggers MAVS-dependent response in tumor cells.** Innate immunity sensors like RIG-I or cGAS are key mediators of innate immune signaling[20] that have recently been shown to induce cell death when activated in cancer cells[15,16,24,48]. To test the effects of direct activation of individual NARs we employed various agonists across kinase-driven cell lines. We found that only activation of RIG-I via IVT4 but not of cGAS by pDNA, TLR7/8 by R848, TLR3 by untransfected poly-(I:C), or MDA5 using TransIT-LT1 transfected poly-(I:C) induced IL-6 (IVT4 $p = 0.04$), CXCL-10 (IVT4 $p = 0.02$) or IFN ($p = 9 \times 10^{-5}$) secretion across the cells (Fig. 4a, Supplementary Fig. 6a,b). While TLR expression was generally low in the cell lines, response to MDA5 and cGAS agonists was limited despite measurable RNA levels (Supplementary Fig. 6c), as described in other tumor models[22]. As expected, RIG-I-mediated secretion of cytokines was dependent on MAVS and not STING expression as confirmed by a CRISPR/Cas9-based knockout of either *MAVS* or *STING* (PC9 e.v. vs. sgMAVS IL-6 $p = 4 \times 10^{-5}$, CXCL10 $p = 2 \times 10^{-12}$) (Fig. 4b, Supplementary Fig. 6d–f). In contrast, the

induction of IFN target genes triggered by osimertinib (Fig. 2a) was comparable between e.v., *MAVS*- and *STING*-knockout cells, further underlining that MAPK- rather than MAVS-/STING-TBK1-IRF3 signaling may be driving this response (Fig. 4c, Supplementary Fig. S6). In line with this, in EGFR-mutant cells, we observed a MAVS-dependent induction of TBK1 phosphorylation only in cells treated with a RIG-I agonist but not in cells treated with osimertinib (Supplementary Fig. S6). At the same time, both IVT4 and inhibition of EGFR/MAPK signaling had a similar effect on the induction of IRF1 but only direct RIG-I activation led to robust PARP cleavage either alone or in combination with osimertinib in PC9 cells after 24 h (Fig. 4d). Using 3′ RNA-seq we observed a massive upregulation of IFNγ target genes, but also the upregulation of apoptosis-related genes in two IVT4 treated cell lines (Fig. 4e). More specifically, it strongly induced pro-apoptotic genes *PUMA* (*BBC3*) and *NOXA* (*PMAIP1I*), which could support IVT4-mediated cell killing (Fig. 4f). Finally, we tested the cell viability effects of direct activation of various innate immunity sensors on a panel of cell lines driven by diverse activated kinases. The ability to reduce cellular viability through activation of innate immune signaling followed the trend for the efficacy to induce cytokine secretion that was strongest with RIG-I activation (IVT4 $p = 0.006$) (Fig. 4g). Again, this effect was dependent on MAVS-expression as determined in PC9 genome-engineered cells (Supplementary Fig. 6i) and lasted for up to 72 h across cell lines (Supplementary Fig. 6j). Thus, in contrast to inhibition of oncogene signaling, direct RIG-I activation leads to a MAVS-dependent induction of cytokine secretion and cell killing across cancer cells.

**Combination of kinase inhibition and RIG-I activation is synergistic in tumors.** We next aimed to investigate the potential of therapeutically exploiting the kinase inhibitor induced inflammatory signaling in combination with direct RIG-I activation. Detection of cell death induction with AnnV/PI staining revealed a synergistic effect of osimertinib and IVT4 in *EGFR*-mutant PC9 cells only when cells expressed MAVS (Fig. 5a), which tracked well with differences in inflammatory target gene induction (Supplementary Fig. 7a). Furthermore, in Colo205 (*BRAF*^mut), A375 (*BRAF*^mut), and A549 (*KRAS*^mut) we observed a significant reduction of cell viability upon IVT4 treatment in cells pre-treated with trametinib or vemurafenib when compared to the respective controls (Fig. 5b). Similarly, the percentage of cell death measured by AnnV/PI-analysis was

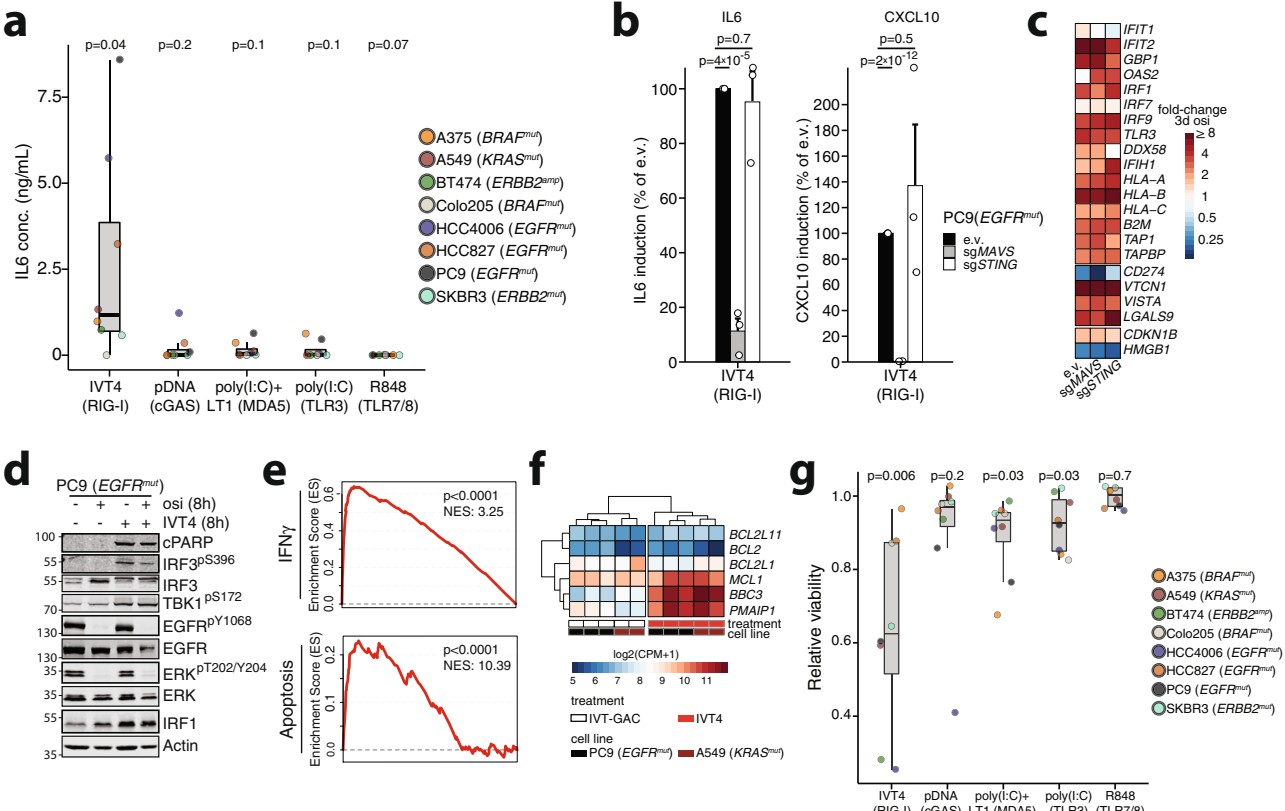

**Fig. 4 Nucleic acid receptor agonists induces cytokine secretion and impairs cell growth in oncogene-driven cancer cells. a** ELISA of IL6 secretion from oncogene-driven cells following stimulation with NAR agonists for 16 h. (points indicate mean of $n = 3$ independent replicates per cell line) Boxplots display median (center line), 25th/75th percentile (lower/upper box hinges), whiskers extend to the most extreme value within 1.5× interquartile range (IQR) of the hinges. **b** Secretion of IL6 and CXCL10 in PC9 cells carrying CRISPRv2 e.v. or with a MAVS or STING sgRNA after 16 h stimulation with RIG-I agonist IVT4 (1 ng/μL) (mean ± SEM of $n = 3$ independent biological replicates). **c** RNA-seq based expression of IFN target genes in cells from (**b**) treated with osimertinib (300 nM, 3d) compared to DMSO controls. **d** Immunoblot following osimertinib or IVT4 treatment (1 ng/μL) (cPARP = cleaved PARP). Representative image of $n = 3$ biological replicates. **e** GSEA analysis of joint differential expression analysis for RNA-seq of PC9 and A549 cells after 8 h IVT4 treatment (1 ng/μL) compared to control (NES = normalized enrichment score, FDR-adjusted $q$-values of Kolmogorov–Smirnov-based permutation test shown). **f** Expression of apoptosis-related genes in IVT4-treated cells from **e** (CPM = count per million). **g** Relative viability of cancer cell lines stimulated with innate immunity agonists for 16 h measured with MTT assay (normalized to controls) (points indicate mean of $n = 3$ independent biological replicates per cell line and agonist). Boxplots display median (center line), 25th/75th percentile (lower/upper box hinges), whiskers extend to the most extreme value within 1.5× interquartile range (IQR) of the hinges. Significance was calculated by one-sample $t$ tests (**a**, **g**) and two-sample $t$ tests (**b**). All tests are two-sided. Source data are provided as a Source Data file.

significantly greater following pre-treatment (Fig. 5c). Similarly, relative viability after IVT4 treatment was observed after pre-treatment with trametinib, but not osimertinib in PC9[T790M+C797S] cells (Supplementary Fig. 7b). Finally, we asked whether forced IRF1 expression alone may suffice to induce a response to IVT4 and indeed observed a moderate, but significant effect in PC9 cells (Fig. 5d). Interestingly, *IRF1* knock-out cells did not show relevant differences in cell death compared to e.v. cells in response to IVT4 (Supplementary Fig. 7c). This prompted us to further investigate the mode of cell death following IVT4 treatment. Using inhibitors of ferroptosis (Fer1), necroptosis (Nec1s), and caspases (ZVAD) indicated that IVT4 primarily induces caspase-dependent cell death across different cell lines (Fig. S7d). It has been described previously that kinase-inhibitor treatment increases susceptibility to apoptosis and higher dependence on anti-apoptotic family members like MCL1[29]. Considering that pro-apoptotic genes NOXA and PUMA are induced by IVT4 (Fig. 4g) we speculated that those may contribute to the observed synergistic effects of osimertinib with IVT4. Indeed, knockdown of *NOXA*, but not *PUMA* significantly reduced the number of apoptotic cells induced by combination treatment (Supplementary Fig. 7e, f).

To test whether we may also exploit the increased sensitivity towards RIG-I agonists after MAPK-dependent reprogramming in vivo we used A549 xenografts and combined trametinib induction treatment with IVT4. In these xenografts we observed considerable tumor shrinkage with trametinib (Supplementary Fig. 8a). However, after the pre-treatment phase tumor shrinkage was larger with IVT4 than with the unspecific control IVT-GAC in trametinib treated ($p = 0.01$), but not in control mice ($p = 0.14$) compared to the start of IVT4/IVT-GAC treatment (Supplementary Fig. 8b). Treatment did not lead to loss of body weight in mice (Supplementary Fig. 8c). Encouraged by these results we next extended the analyses to our humanized PC9 xenografts that offer the presence of immune cells. In this setting, treatment duration is more limited due to the risk of graft-versus-host disease. Osimertinib or vehicle treatment was therefore given in a run-in fashion and then stopped to assess IVT4 effects during tumor regrowth. As expected, osimertinib immediately reduced tumor growth compared to vehicle control (Fig. 5e). However, after stopping osimertinib injection of IVT4 not only inhibited outgrowth of the tumor compared to the non-specific control IVT-GAC but led to a further reduction in tumor volume and a

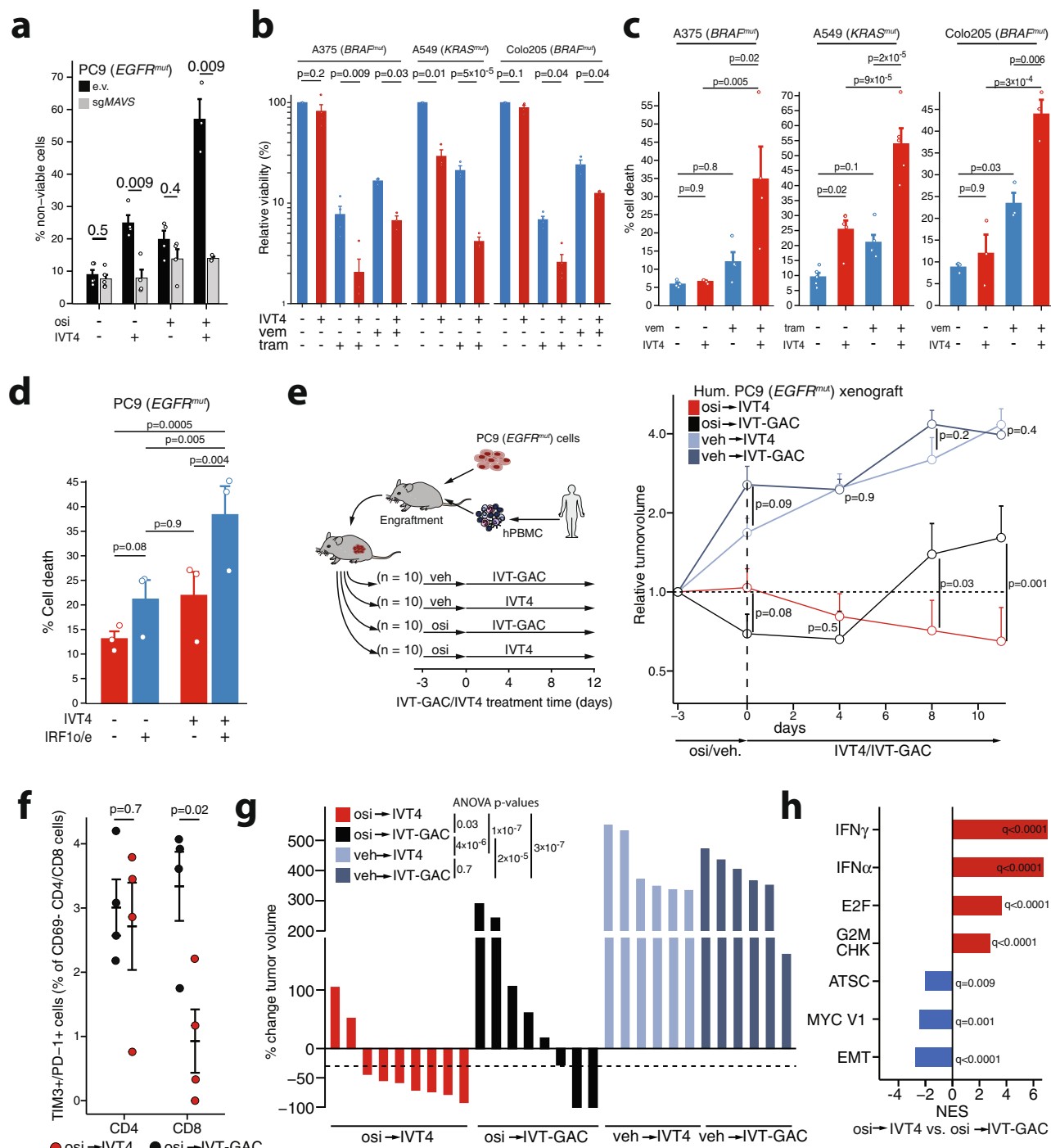

lower rate of TIM3+/PD-1+CD69− CD8 cells with osi pre-treatment (Fig. 5f, g, Supplementary Fig. 8d, e). Without osimertinib pre-treatment, no major changes in tumor volume were observed between IVT4 and the IVT-GAC arms (Fig. 5e, g). Similarly, tumor outgrowth was comparable between IVT4 and IVT-GAC in a non-humanized PC9 xenograft irrespective of osimertinib pre-treatment, indicating a relevant induction of an immune response by the combination treatment in the humanized setting (Supplementary Fig. 8f).

In contrast, pre-treatment with osimertinib did not have a major effect on the group of mice treated with a PD-L1 inhibitor (Supplementary Fig. 8g, h). Investigating the transcriptional profiles in the IVT4/IVT-GAC cohorts revealed induction of IFN-target genes in IVT4-treated tumors with or without

osimertinib pre-treatment compared to the respective IVT-GAC controls (Fig. 5h, Supplementary Fig. 8i). However, among tumors pre-treated with osimertinib, osi/IVT-GAC ($n = 6$) tumors showed a stronger enrichment of EMT and ATSC gene sets compared to osi/IVT4-treated tumors ($n = 6$) (Figs. 1a and 5h). This differential enrichment of the ATSC/EMT gene expression profiles in osi/IVT4 vs. osi/IVT-GAC treatment may be associated with the selection pressure of IVT4 treatment in vivo. Accordingly, IFN target genes with increased VTCN1 levels were lower overall in osimertinib/IVT-GAC-treated tumors (Supplementary Fig. 8j). Overall, these results show that the combination of oncogene inhibition may induce a MAPK-dependent transcriptional reprogramming that sensitizes lung cancer xenograft towards RIG-I agonists.

**Fig. 5 Targeted kinase-inhibition enhances NAR agonist-induced cell death. a** Flow-cytometric analysis of cell death induction following 24 h treatment of PC9 cells carrying CRISPRv2 e.v. or sgMAVS with DMSO/osimertinib (300 nM) and IVT4/IVT-GAC (1 ng/µL) (vertical axis displays the normalized percentage of AnnexinV and/or PI-positive cells = % non-viable cells) (mean ± SEM of independent experiments with $n = 3$ osi + IVT4, $n = 4$ other treatments). **b** CTG assay for cell lines pre-treated (48 h) with trametinib (tram, 100 nM) or vemurafenib (vem, 1 µM) and subsequent addition of IVT4/ IVT-GAC (1 ng/µL) for 24 h. Viability was normalized to respective IVT-GAC controls (mean ± SEM log10 viability of independent experiments: $n = 3$ A549, vem A375, Colo205, and $n = 4$ otherwise). **c** Flow-cytometric analysis of cell death induction for cell lines pre-treated (48 h) with trametinib (100 nM) or vemurafenib (1 µM) and subsequent addition of IVT4/IVT-GAC (1 ng/µL for 24 h (vertical axis displays the normalized percentage of AnnexinV and/or PI-positive cells = % cell death) (mean ± SEM of independent biological replicates for A375 $n = 4$, A549 $n = 5$, Colo205 $n = 3$). **d** Flow-cytometric analysis of cell death induction in PC9 cells transfected for 48 h with IRF1 or e.v. followed by the addition of IVT4/IVT-GAC (1 ng/µL) for 24 h (vertical axis displays the percentage of AnnexinV and/or PI-positive cells = Cell death %) (mean ± SEM of $n = 3$ independent experiments). **e** Humanized PC9 xenografts treated with osimertinib or vehicle p.o. followed by IVT4 or control IVT-GAC i.t. as shown in the schematic (left, $n = 5$ mice per arm inoculated with 2 tumors per mouse). Relative tumor volumes are shown on the right (mean ± SEM of tumors treated with osi-IVT4 ($n = 9$), osi-IVT-GAC ($n = 8$), veh-IVT4 ($n = 6$), veh-IVT-GAC ($n = 6$)). **f** Flow-cytometry of tumor-infiltrating CD4 and CD8 lymphocytes for TIM3 and PD1 expression after 4 days osimertinib and 6 days IVT4/IVT-GAC. (each data point = one of $n = 4$ tumors per group, error bars represent mean ± SEM). **g** Tumor volumes at study end compared to treatment start in mice from (**e**) (each bar = one tumor). **h** GSEA on RNA-seq of xenograft tumors in humanized mice after pre-treatment with osimertinib and IVT4/IVT-GAC. (FDR-adj. $q$-values). Significance was calculated by **t** tests (**a**, **e**) and paired $t$ tests (**b**) adjusted for multiple testing with Bonferroni–Holm method; ANOVA with Tukey post-hoc tests (**c**, **f**), two-way ANOVA adjusting for the mouse in osi or vehicle groups (**f**) and two-way ANOVA with Tukey post hoc tests (**d**, **g**). All tests are two-sided. Source data are provided as a Source Data file.

**Full anti-tumor activity of combinatorial IVT4 and kinase inhibition requires CD8 and NK cells.** We next aimed to validate our data in a fully immunocompetent mouse model to assess the tumor intrinsic and immune-mediated effects following kinase inhibition and upon IVT4 treatment. To this end, we employed an *Egfr*-mutant, syngeneic mouse model, which allows longer-term observation and thus a combination of IVT4 during continuous osimertinib treatment[49,50]. In line with our cellular data, osimertinib monotherapy led to reduced tumor growth and increased expression of inflammatory genes also in this mouse model (Fig. 6a, b, Supplementary Fig. 9a). Furthermore, following an osimertinib lead-in, the combination of osimertinib with IVT4 induced significant tumor shrinkage when compared to osimertinib combined with the unspecific control IVT-GAC (Fig. 6c, d). The combination treatment was well tolerated and not associated with altered splenic CD4/CD8 ratio or bodyweight loss in these mice (Supplementary Fig. 9b, c). To assess the IVT4-induced changes in the microenvironment we analyzed tumors in mice treated with osimertinib and IVT4 or IVT-GAC by flow-cytometry. While the number of NK and CD8 cells were comparable between both groups, IVT4 treatment significantly reduced PD-1 expression in tumor-infiltrating CD8 cells (Supplementary Fig. 9d).

Prior studies have shown that anti-tumor efficacy of RIG-I agonists is enhanced by NK or T cell-mediated effects[24,51]. Since we observed direct effects of IVT4 on tumor cells in vitro including cytokine secretion and cell death induction, we next aimed to assess the contribution of tumor-cell intrinsic and tumor micro-environment-related effects. Following the encouraging first treatment cohorts, we thus repeated the study in our syngeneic mouse model but also included CD8-depleted or NK-cell-depleted treatment arms (Supplementary Fig. 9e, f). Interestingly, early after starting IVT4 treatment tumor size reductions in all osimertinib-IVT4 arms were comparable (Fig. 6e). However, during prolonged treatment tumor shrinkage was significantly greater in the non-depleted osimertinib-IVT4-IgG arm compared to the arms with CD8-cell or NK-cell depletion (Fig. 6e, f, Supplementary Fig. 9e). In the vehicle-treated arms, no differences in tumor volume were evident with depletion of CD8 or NK-cells (Supplementary Fig. 9e). This suggests that the effects of both immune cell populations may contribute to the delayed effects of IVT4 in an in vivo setting and may further enhance the direct tumor-cell intrinsic effects (Fig. 6g).

## Discussion

Over the past decade, the therapeutic options for patients with oncogene-driven tumors have expanded considerably and personalized therapies significantly prolong patient survival in genetically defined subgroups[1–3]. Nevertheless, virtually all responders develop resistance after a phase of response in which tumor cells remain viable despite continuous drug exposure. This clinical observation is in line with the notion that drug resistance rapidly emerges even in simple cell culture monolayer models of various cancer types[4,6,7,9,10]. In these models drug, persistent cells remain viable in a quiescent state of limited proliferative capacity[4,6–8,11,12]. This phenotype in part resembles the cell-cycle arrest program observed in senescent cells during therapeutic stress[12,28]. However, the stress-responsive reprogramming of persistent cells, especially within the context of the tumor microenvironment lacks a mechanistic framework.

To address this issue, we explored the global effects of targeted inhibition of oncogenic signaling across time and different tumor types. Here, we uncover an unexpected mechanistic link between MAPK signaling and the control of inflammatory networks during therapy-induced reprogramming of tumors. Across a broad panel of oncogene-driven pre-clinical models, we observed a highly conserved transcriptomic signature following kinase inhibition including induction of IFN-target genes, which was most pronounced in EGFR-driven and BRAF-driven cancers. We also observed comparable patterns for BRAF-mutant melanoma and HER2-positive breast cancer patients. Of note, the immunological effects mediated by the monoclonal antibody trastuzumab, which was used as a targeted therapy in the breast cancer cohort, may also add to the observed effects. Taken together, MAPK signaling may be actively involved in the fine-tuning of the tumor immune-environment across cancer types.

Similar transcriptional profiles have been described following treatment of tumor cells with epigenetic drugs[15,16] or CDK-inhibitors[11,13,14,17]. Of importance, our data suggest that similar effects are inducible via inhibition of the oncogenic driver and its downstream MAPK signaling independently of cytotoxic effects. Limited induction of IRF1 or IFIT1 was also observed following PI3K-inhibition in some cell lines, potentially due to cross-pathway regulation or cell-intrinsic differences. However, across various genotypes, this reaction appears to be driven by the MAPK-mediated de-repression of IRF1 and subsequent upregulation of IFN-response genes as indicated by RNA and ChIP data. Surprisingly, knock-out of the IFN receptor *IFNAR1* did not impede induction of IFIT1 or IRF1 by osimertinib treatment.

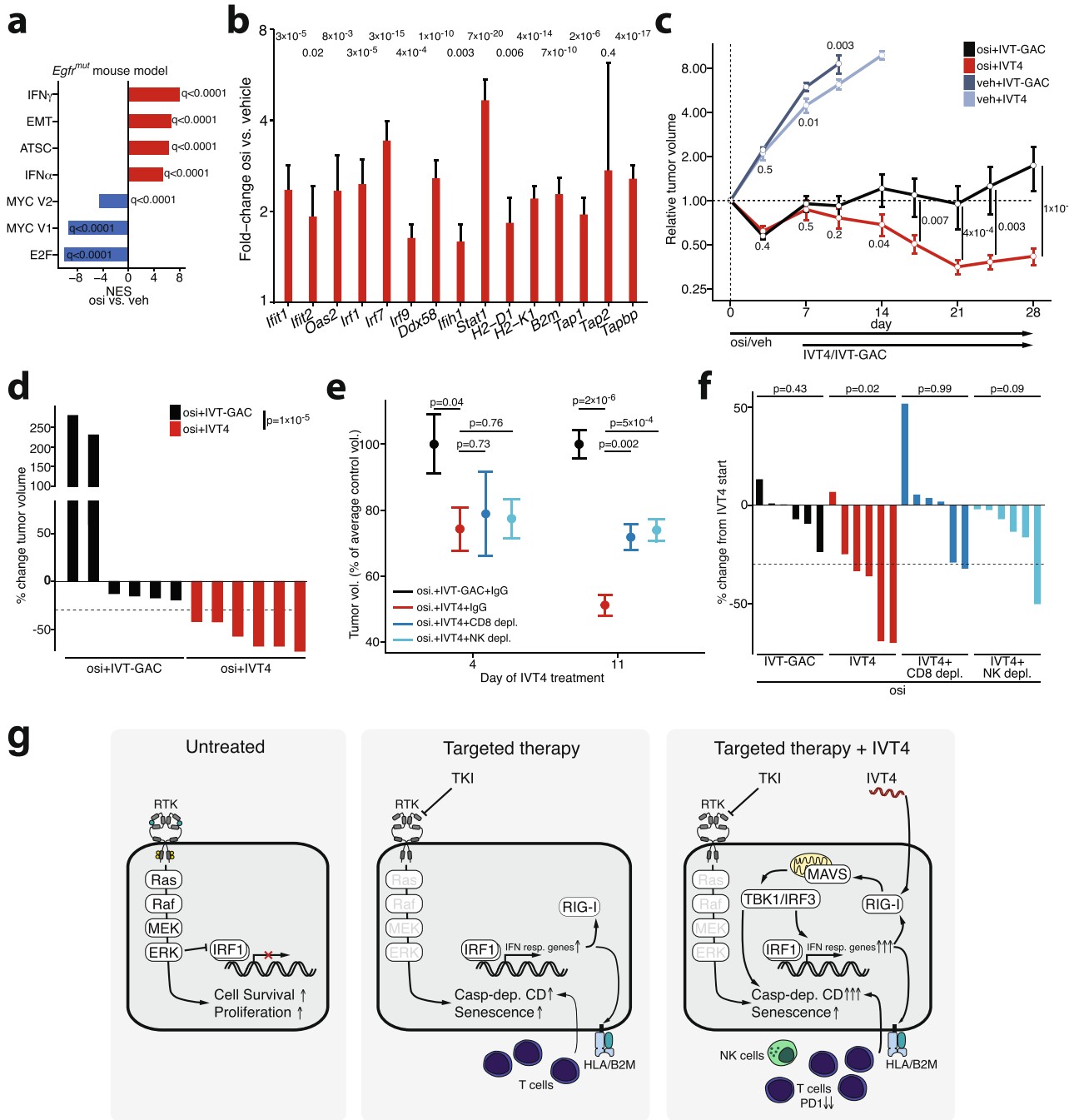

**Fig. 6 Combined kinase inhibitor and RIG-I agonist treatment in an EGFR^mut syngeneic mouse model. a** RNA-seq followed by GSEA of *Egfr^mut* syngenic mouse models comparing all control mice (0 days, 10 days vehicle; *n* = 4) vs. all osi treated mice (4 days, 10 days; *n* = 4). Two mice per treatment time-point, one tumor per mouse analyzed (in total *n* = 8 mice). **b** RNA-seq analysis of inflammatory gene s for mice from (**a**) comparing all control (0 days, 10 days vehicle) vs. all osi-treated mice (bar height indicates fold-change from differential expression analysis, error bar: standard error of the fold-change; FDR-adjusted *q*-values shown). **c** Relative tumor volumes of syngenic *Egfr^mut* mice treated with osimertinib or vehicle p.o. subsequent addition of IVT4 or IVT-GAC i.t. (mean ± SEM of *n* = 6 tumors per group). **d** Change of individual tumor volumes in mice from (**c**). **e** Relative tumor volumes of syngenic *Egfr^mut* mice treated with osimertinib and IVT4/IVT-GAC with the addition of depleting antibodies for CD8 or NK cells or IgG control. Volumes were normalized to the average of the control group per time point (mean ± SEM of *n* = 6 tumors per group). **f** Individual tumor volumes from (**f**) at tumor harvest displayed as percent of their volume at the start of IVT4/IVT-GAC (one bar = one tumor, max. 2 per mouse). **g** Schematic of the proposed processes driving the response to tyrosine kinase inhibition (TKI) and IVT4 treatment (Casp-dep. CD = Caspase-dependent cell death, RTK = receptor tyrosine kinase). Significance was calculated as FDR-adjusted *q*-values (**a**, **b**), by two-way ANOVA adjusting for the mouse in osi or vehicle groups (**c**, **d**), two-sample *t* tests (**e**), and one-sample *t* tests (**f**). All tests are two-sided. Source data are provided as a Source Data file.

Together with the lack of detectable IFN in the medium supernatant, this suggests a cell-autonomous process. Considering the time dynamics, the lack of robust IFN secretion by treated cells, the limited engagement of the TBK1–IRF3 axis, and the sustained activation following MAVS knock-out indicate a tumor cell-autonomous process distinct from previously described mechanisms[52]. It has also been shown that cell death can modify and engage inflammatory programs[53]. However, our in vitro data using caspase-inhibitors to prevent cell death following kinase inhibition suggests that this is not likely a major driver of the observed effects. Our data rather suggest that suppression of MAPK activity induces a de novo transcription of IRF1 and of other IRF family members leading to a TBK1/IRF3-independent inflammatory reprogramming.

Similar drug-induced inflammatory response profiles have been suggested to enhance ICB efficacy[14,16]. In general, ICB has shown relevant benefits across many cancer entities, however in several oncogene-driven tumors responses have been disappointing, e.g. in ALK- or EGFR-driven lung cancer[54,55], where ICB may even be related with severely increased adverse event rates in certain scenarios[56]. In accordance with previous reports, PD-L1 expression depended upon EGFR activity[57] and PD-1/EGFR inhibitor combination could not prevent tumor outgrowth in our humanized PDX models. At the same time, we observed an upregulation of alternative immune checkpoints such as VTCN1 (B7-H4), which has been associated with limited ICB response and reduced T-cell activity previously[41,58]. Of note, upregulation of VTCN1 was not observed in BRAF-mutant melanoma cells or patient samples, indicating additional levels of regulation.

Among the genes induced during inflammatory reprogramming, we also noticed several members of the innate immune system including NARs like RIG-I (DDX58). Activation of NARs in tumor cells by intrinsic and extrinsic agonists has been shown previously to invoke anti-tumor responses[16,24,48,51]. Among the tested NAR agonists, IVT4 activating RIG-I was able to induce an IFN response across the cell lines and induced TBK1-IRF3 signaling. This inflammatory response was accompanied by the upregulation of pro-apoptotic genes such as PUMA and NOXA and resulted in cytotoxicity across different cell lines. More importantly, in combination with targeted treatment IVT4 was able to significantly increase cell death in a MAVS-dependent fashion in vitro. These findings also translated into robust tumor shrinkage of immune-competent EGFR-driven in vivo models when IVT4 was combined with the EGFR inhibitor osimertinib. Furthermore, the delayed in vivo effects of RIG-I activation involved the presence of NK- and CD8 cells. Moreover, both in the humanized xenograft model as well as in the syngeneic mice IVT4 treatment reduced levels of exhausted CD8 cells, which further supports immune activation as a mechanism leading to the efficacy of combined kinase inhibition with RIG-I stimulation.

Thus, while the single-agent kinase inhibitor treatment promotes the recruitment of tumor-infiltrating lymphocytes only the combination with a RIG-I agonist effectively promotes the clearance of drug-tolerant persisters. Currently, several agonists of innate immunity sensors including RIG-I are evaluated in clinical trials[59], and novel RIG-I agonists that can be administered orally[60] or intravenously[26] will help to broaden the scope of clinical translation.

Taken together, we propose that the drug-induced reprogramming of innate immunity signaling can be exploited using RIG-I agonists to maximize the anti-tumor efficacy of targeted tumor treatment. Together with the observed induction of alternative immune checkpoints our findings thus not only offer insights into the adaptation processes following kinase inhibition but provide promising novel therapeutic avenues for oncogene-driven cancers.

## Methods

**Cell culture**. Human cancer cell lines were obtained from ATCC and verified regularly for Mycoplasma infection and by short tandem repeat (STR) profiling at the Institute for Forensic Medicine of the University Hospital of Cologne. PC9, HCC827, HCC4006, BT-474, H3122, A375, and H1993 cells were cultured in RPMI; Colo205, A375, SKBR3, and HEK293T cells were cultured in DMEM. All media were supplemented with 10% fetal bovine serum and 1% penicillin/streptomycin. All cells were grown at 37 °C in a humidified atmosphere with 5% $CO_2$. Primary murine PDAC cell lines had been derived previously from tumor pieces of $Ptf1a^{wt/Cre}$; $Kras^{wt/LSL-G12D}$; $Trp53^{loxP/loxP}$ mice[38] through incubation in DMEM high-glucose medium (Thermo Fisher Scientific) supplemented with 10% fetal bovine serum (Thermo Fisher Scientific) until proliferating tumor cells could be established.

**Immunoblot**. Cells were washed with PBS and proteins were extracted in RIPA buffer with protease inhibitors cOmplete Mini Protease Inhibitor Cocktail (Roche) and phenylmethylsulfonylfluorid (0.1 mM). Protein concentrations were determined by BCA assay (Pierce). Equal amounts of protein were separated on 4–20% Tris–glycine sodium dodecyl sulfate–polyacrylamide gel electrophoresis (SDS–PAGE) gels (Thermo Fisher Scientific, USA) and transferred to PVDF-FL membranes (Millipore). Upon blocking, membranes were stained with primary antibodies overnight, washed, and incubated with fluorescently labeled secondary antibodies. Fluorescence was detected with the Odyssey CLx imaging system (LI-COR Biosciences, USA).

**Cell viability screening**. Cells were seeded as triplicates in 96-well plates (Costar) and serially diluted drug or DMSO was added the next day. For PDAC cell lines trametinib was pre-printed using the D300e Digital Dispenser (Tecan, Switzerland). Viability was determined after 72 h by CellTiter-Glo assay (CTG) (Promega) in an Infinite 200 Pro microplate reader (Tecan, Switzerland) for humans and a Tecan Spark 10M multiple readers for PDAC cell lines. Data were averaged across replicates and dose–response curves were fitted using R or GraphPad Prism to infer half-maximal growth-inhibitory ($GI_{50}$) concentrations.

**Viability analysis of combined kinase inhibition and IVT4 treatment**. Cells were seeded in 96-well plates and treated with 100 nM trametinib, 300 nM osimertinib, 1 μM vemurafenib, or DMSO the next day. After 48 h of treatment, IVT4/IVT-GAC were mixed in Opti-MEM (ThermoFisher, USA) with Lipofectamine 2000 (Invitrogen, USA) and added to achieve a final concentration of 1 ng/μL IVT4. The next day cell viability was assessed using CellTiter-Glo assay (CTG) (Promega) in an Infinite 2000 Pro microplate reader (Tecan, Switzerland). Values were normalized to the respective IVT-GAC controls per compound and cell line.

**RNA sequencing**. Transcriptome sequencing was generated during this study for patient-derived cell lines and samples and the lung adenocarcinoma mouse models were performed using total extracted RNA. 3′ UTR mRNA libraries were generated from total RNA using the Lexogen QuantSeq kit (Lexogen, Austria) according to the standard protocol and sequenced on Illumina HiSeq4000 or NovaSeq sequencer (Illumina, USA). RNA extracted from primary murine PDAC cell lines was processed using TruSeq Stranded mRNA Kit (Illumina, USA) and sequenced as 2 × 100 bp on a HiSeq 4000 (Illumina, USA). Raw-sequencing data were aligned to the respective reference genomes and quantified prior to differential expression analyses. For details see Supplementary Materials and Methods.

**Humanized mouse model**. PBMCs were isolated from healthy donor buffy coats (ethics approval 18-198, University of Cologne) by Ficoll-Paque density gradient (GE Healthcare, USA) centrifugation and used without intermittent freezing. 8–10-week-old NSG mice (NOD.Cg-Prkdcscid Il2rgtm1Wjl/SzJ) were injected s.c. with $5 \times 10^6$ PC9 cells in both flanks and $10 \times 10^6$ healthy donors PBMCs i.p. Tumor growth was monitored 2×/week and treatment was commenced on day 17 after tumor inoculation and PBMC engraftment (Animal study approval number 84-02.04.2017.A236). The tumor size was approximately 150 mm³ on day 17 when treatment commenced. For treatment details see Supplementary Materials and methods.

**Syngeneic $Egfr^{mut}$ mouse model**. Female C57BL/6J mice were purchased from Charles River Laboratories Japan, Inc. (Yokohama, Japan). All mice were provided sterilized food and water and were maintained at an air-conditioned temperature of 22 ± 2 °C with constant humidity under a 12/12-h light/dark cycle. Murine Egfr mutant adenocarcinoma tumors[45] were dissociated into single-cell suspensions by using a Tumor Dissociation Kit, mouse (Miltenyi Biotec, Germany), red blood cells were removed from the suspensions (Red Blood Cell Lysis Solution; Miltenyi Biotec), 50–100 × 10⁴ cells were resuspended in 0.1 mL PBS and 0.1 mL of Matrigel matrix (Corning, New York, USA), and injected subcutaneously into the back (on both sides) of C57BL/6J mice. Treatment with osimertinib and IVT4/IVT-GAC was performed as described in the Supplementary Materials and methods.

**Statistical analyses**. Statistical analyses were performed as described in the figure legend for each experiment. Data are expressed as mean ± SEM with significance set at $p < 0.05$ unless otherwise noted. The sample size was not predetermined. All samples meeting proper experimental conditions were included in analyses. Box-plots: Box edges display 25th and 75th percentile, middle lines the median and whiskers extend to the value max. 1.5*IQR of the box edges (IQR = interquartile range). Points beyond 1.5*IQR are drawn individually. Statistical significance was determined as indicated in the figure legends using two-tailed tests using Prism software (v9.1, GraphPad Software, USA) or the statistical environment R (v3.5.0).

Primers used for RT-qPCR, CRISPR experiments, ChIP analyses, and DNA sequencing can be found in Supplementary Table S2, gating strategies in Supplementary Fig. S10. All experiments including animal studies were performed in agreement with local regulations by trained researchers. Animal experiments were approved by the Institutional Animal Welfare Committees of the University of Cologne (84-02.04.2017.A236) and the University of Okayama (OKU-2017412). Experiments with human material approved by Crown Bioscience and the University of Cologne (ethics approval 18-198). For additional details please see Supplementary Materials and methods.

**Reporting summary**. Further information on research design is available in the Nature Research Reporting Summary linked to this article.

## Data availability

The RNA-seq data of murine PDAC cells generated in this study have been deposited in the Gene Expression Omnibus database under accession code GSE181599 (www.ncbi.nlm.nih.gov/geo/). All other sequencing data has been deposited in the ArrayExpress database under accession codes E-MTAB-9468, E-MTAB-9883, E-MTAB-9884, E-MTAB-9885, E-MTAB-9887, E-MTAB-9888, E-MTAB-9889, E-MTAB-9890, E-MTAB9891, E-MTAB-9886, E-MTAB-10851 and E-MTAB10802. We also used the publicly accessible data sets GSE76360 (breast cancer RNA expression), GSE100336 (*KRAS* G12C expression data) (www.ncbi.nlm.nih.gov/geo/) and EGAS00001000992 (melanoma patient RNA-seq), EGAS00001002335 (LC2/AD RNA-seq) (https://ega-archive.org). The remaining data are available within the Article, Supplementary Information or Source Data file. Source data are provided with this paper.

## Materials availability

Further information and requests for resources and reagents should be directed to and will be fulfilled upon reasonable request by the Lead Contact, Martin Sos. Source data are provided with this paper.

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

## Acknowledgements

The CRISPR-engineered Lewis Lung Carcinoma cell lines were a kind gift from Julian Downward. lentiCRISPR v2 was a gift from Feng Zhang (Addgene plasmid # 52961). The authors thank CeGaT GmbH (Tübingen) for RNA-sequencing of murine PDAC cell lines. This work was supported by the Deutsche Krebshilfe (70113009 to R.T.U.; 70113129 to C.L.; 70112888 to M.L.S., 70112505 and 70113835 to J.T.S., Max-Eder-Junior Research Group 701125509 to S.vK., Mildred Scheel Nachwuchszentrum Grant 70113307 to D.A. and J.B.), by the Nachwuchsforschungsgruppen-NRW (1411ng005 to R.T.U.), the Japan Society for the Promotion of Science(JSPS) (Grant-in-Aid for Scientific Research (B) 19K08625 to K.O.), the European Union (Erasmus Fellowship to E.S.T.), the Else Kröner Fresenius Stiftung (Memorial Grant 2018_EKMS.35 to J.B.; 2016-Kolleg-19 to S.K. and S.B.), the Frauke Weiskam + Christel Ruranski Foundation (S.B.), German Research Foundation (DFG) (PL 894/1-1 to D.P.; SCHL1930/1-2, TRR237; Excellence Cluster ImmunoSensation to M.S. and G.H., SFB670 to G.H., CRC1399 Project-ID 413326622 to M.L.S., R.K.T., S.vK., J.G., and R.B., SI1549/3-1 (KFO337; PhenoTime) to J.T.S.), the German Cancer Consortium (DKTK) to J.T.S., and the Bundesministerium für Bildung und Forschung (eMed consortium grant 01ZX1901A to S.vK., R.K.T., J.G., and M.L.S.; 01ZX1406 to M.L.S.), by the Fritz Thyssen Foundation (10.19.2.025MN to M.L.S.) and the German federal state North Rhine Westphalia (NRW) as part of the EFRE initiative (EFRE-0800397 to M.L.S., R.K.T., and R.B.).

## Author contributions

Conceptualization: J.B., M.L.S.; Investigation: J.B., C.L., S.B., K.N., J.W., J.O., D.F.A., A.H., D.P., S.K., P.L., T.Z., S.L., E.S.T., M.A.D., L.M., M.H., A.Q., J.D. T.N., A.Hi., S.O., P.L.; Methodology: J.B., C.L., D.F.A., D.P., T.Z., K.N., L.K.G., J.G., J.F., M.T., K.O., R.T.U., Q.L., S.B., S.K.; Formal analysis: J.B., C.L.; Resources: S.vK., M.S., G.H., R.B., J.T.S., Q.L., R.T.U., M.L.S.; Writing—original draft: J.B., C.L., M.L.S.; Writing—Review & editing: J.B., J.G., R.K.T., C.L., A.Q., S.B. K.N., J.W., J.O., D.P., S.vK., S.K. S.B., R.T.U., R.B., M.S., G.H., L.K.G., J.T.S., K.O., C.A.S., M.L.S.; Supervision: M.L.S.

## Funding

## Competing interests

M.L.S. is a founder and shareholder of PearlRiver Bio (now part of Centessa Pharmaceuticals) and received consulting honoraria from PearlRiver Bio. M.L.S. receives research funding from PearlRiver Bio and Novartis. R.B. is an employee of Targos Molecular Pathology. H.L. is an employee of CrownBiosciences. G.H. is co-founder of Rigontec GmbH. M.S. is listed as inventor on a patent application covering RIG-I activating structures. R.K.T. is founder of PearlRiver Bio (now part of Centessa Pharmaceuticals), founder of NEO New Oncology (now part of Siemens Healthcare), consulting honoraria from PearlRiver Bio and NEO New Oncology. K.O. received research funding from Boehringer Ingelheim, Novartis, AstraZeneca, Eli Lilly, and Daiichi-Sankyo outside the submitted work. K.O. reports honoraria from AstraZeneca, MSD and Chugai pharmaceutical outside the submitted work. The other authors declare no competing interests.

## Additional information

[1]Molecular Pathology, Institute of Pathology, Faculty of Medicine and University Hospital Cologne, University of Cologne, 50931 Cologne, Germany. [2]Department of Translational Genomics, Faculty of Medicine and University Hospital Cologne, University of Cologne, 50931 Cologne, Germany. [3]Center for Molecular Medicine Cologne, Faculty of Medicine and University Hospital Cologne, University of Cologne, 50931 Cologne, Germany. [4]Mildred Scheel School of Oncology Cologne, Faculty of Medicine and University Hospital Cologne, University of Cologne, 50931 Cologne, Germany. [5]Department I of Internal Medicine, Center for Integrated Oncology Aachen Bonn Cologne Duesseldorf, Faculty of Medicine and University Hospital Cologne, University of Cologne, 50931 Cologne, Germany. [6]Else-Kröner-Forschungskolleg Clonal Evolution in Cancer, Faculty of Medicine and University Hospital Cologne, University of Cologne, 50931 Cologne, Germany. [7]Department of Hematology, Oncology and Respiratory Medicine, Okayama University Graduate School of Medicine, Dentistry and Pharmaceutical Sciences, Okayama, Japan. [8]Institute of Clinical Chemistry and Clinical Pharmacology, University Hospital Bonn, Bonn, Germany. [9]Cold Spring Harbor Laboratory, Cold Spring Harbor, NY 11724, USA. [10]Lustgarten Foundation Pancreatic Cancer Research Laboratory, Cold Spring Harbor, NY 11724, USA. [11]Institute of Pathology, Faculty of Medicine and University Hospital Cologne, University of Cologne, 50931 Cologne, Germany. [12]Institute for Developmental Cancer Therapeutics, West German Cancer Center, University Hospital Essen, Essen, Germany. [13]Division of Solid Tumor Translational Oncology, German Cancer Consortium (DKTK, partner site Essen) and German Cancer Research Center, DKFZ, Heidelberg, Germany. [14]Genome Informatics, Institute of Human Genetics, University Duisburg-Essen, Essen, Germany. [15]Cologne Excellence Cluster on Cellular Stress Response

in Aging-Associated Diseases (CECAD), Faculty of Medicine and University Hospital Cologne, University of Cologne, 50931 Cologne, Germany. [16]Imperial College London, London, UK. [17]Crown Bioscience, San Diego, CA, USA. [18]Department of Hematology, Oncology and Tumor Immunology, Charité - University Medical Center, Virchow Campus, and Molekulares Krebsforschungszentrum, Berlin, Germany. [19]Max-Delbrück-Center for Molecular Medicine in the Helmholtz Association, Berlin, Germany. [20]Department of Hematology and Oncology, Kepler University Hospital, Johannes Kepler University, Linz, Austria. [21]Department of Head and Neck Surgery, Faculty of Medicine and University Hospital Cologne, University of Cologne, 50931 Cologne, Germany. [22]German Cancer Research Center, German Cancer Consortium (DKTK), Heidelberg, Germany. [23]Department of Respiratory Medicine, Okayama University Hospital, Japan, 2-5-1 Shikata-choKitaku Okayama 700-8558, Japan. [24]These authors contributed equally: Johannes Brägelmann, Carina Lorenz. ✉email: johannes.braegelmann@uni-koeln.de; martin.sos@uni-koeln.de

