## [Peer Review File · Nature Communications]

Reviewers' Comments:

Reviewer #1:

Remarks to the Author:

MAPK-mediated inflammatory reprogramming sensitizes tumors to targeted activation of innate immunity sensor RIG-I
By Brägelmann et al.

In the manuscript, the authors uncovered that targeted inhibition of oncogenic signaling in different tumor types induced stemness- and senescence-associated reprogramming, featured as MAPK/IRF1-mediated inflammatory response in tumors. RIG-I agonist IVT4 can be used to maximize the anti-tumor efficacy of targeted therapy. The in vitro results were translated into mice models and achieved solid effect. The experiments are well designed and most of data are reliable. Furthermore, the idea of combining kinase inhibitors with RIG-I agonists against solid cancers is tempting. Nonetheless, substantial revision of the manuscript is required to support their claims.

Introduction

The description regarding target therapy and drug resistance in the first part of discussion should be included in the introduction section. The effect of tumor-intrinsic activation of innate immunity networks should be briefly introduced, not only induction of cell death and immune recognition, but also their role in promoting immunotherapies (doi: 10.1126/sciimmunol.aau8943; doi: 10.1172/JCI131572, doi: 10.1016/j.celrep.2020.108080). This brings their findings with kinase inhibitors and RIG-I agonists in the relevant clinical context.

Results

Figure 1:

In Fig. 1b and Fig. S1F, why no immunoblot band can be seen for p21? All three tested cells are p21 negative? What does "TKI" stands for? Osimertinib? Same labeling should be used. What is the the oncogenic driver for PC9?. In Fig. S1H, legends for the color coding are missing. "higher levels of ... histone marks H3K9me3 and H3K27me3" is not well supported by Fig. S1I, since the loading control is not even, a quantitative immunoblot figure is necessary. In Fig. S1I and J, is there a reason that proteins from different drug treatment time points were used for immunoblotting? For the humanized mouse model, Fig. 1f clearly showed T cells infiltration in the tumor treated with Osimertinib, however, no tumor volume change under Osimertinib treatment is shown. Thus, we have no idea if those infiltrating T cells are functional. What are the phenotypes of these T cells? In Fig.1h, it would be better to show the figure for anti-HER2 base vs. post neoadjuvant therapy, if applicable. Vice versa and to be consistent, Figure 1i and 1j should be divided in responders and non-responders, if applicable.

Figure 2: Fig. 2b/S3a: Did the patients "on" respond? Please provide explanation. Fig. 2C: what is the consequence of these findings? Did these mice respond better than the untreated controls? This is important to assign the clinical relevance of these findings. In Fig. S3D, it is not reasonable to set the y-axis value as 500 (U/ml) without a positive control (e.g. IVTA4 treated cells), the detection method might not work in this experiment. In the figure legends, D was marked as C. In Fig. S3 E and F, why are different loading controls used for immunoblotting of the same cell line? Fig. 2e showed that inhibitor treatment lead to upregulation of VTCN1 (B7-H4), which was previously linked with resistance to immune checkpoint blockade (ICB). In this context, it would be very interesting to demonstrate the phenotype of t infiltrated T cells demonstrated in Fig. 1f, to see the impact of inhibitor treatment on T differentiation and especially if they express an "exhausted" T cell phenotype. What about PD-L1 expression in the tumor cells treated with those inhibitors in Figure 2e, given that type I interferon signaling induces PD-L1 expression. The introduction of DUSP6 and SPRY4 as "two negative feedback regulators of MAPK signaling" somewhat detracts from the main story and causes confusion. I suggest to move the DUSP6 data into supplement. Moreover, why is the x-axis labeling in Fig 2h and 2i different? And also, for Fig. S3N, x-axis and y-axis labelings are different from others. Again, do these results reflect the patient outcome?

Figure 3: "This data indicates that inhibition of the MAPK pathway and not indirect cell death signaling ..." What does "indirect cell death signaling" mean here? In Fig. 3h, the authors showed that IRF1 knock-out significantly reduced transcriptional induction of several IFN target genes. What about the effect of IRF-1 knockout on cell death induced by the inhibitors?

Figure 4: The concentration of IVT4 used in all the experiments of Figure 5 was not provided in the figure or figure legend. Figure 4b: is there a transcriptional profile as in c and cell viability/death data? "While TLR expression was generally low in the cell lines, response to MDA5 and cGAS agonists was limited despite measurable RNA levels (Fig. S5d)." Wrong supplementary figure was cited here. Page 15. Line 24: at this point, it should be stated that these findings are in line with a recent report by Heidegger et al, (doi: 10.1126/sciimmunol.aau8943) where RIG-I agonist seemed more potent than cGAS/STING agonists. For STING-knockout PC9 cells, an immunoblot figure is required to confirm the knockout result. In Fig. 4f, the relative viability of PC9 cell decreased to 0.6 after 16 hours IVT4 treatment, thus, an additional figure of longtime treatment (up to three days) would be interesting. Furthermore, why do some of the oncogene-driven cell lines not respond (e.g. A375, Colo205) to IVT4? Are these the BRAFmut celllines as in in figure 5 and if yes, does that interfere with RIG-I signaling?

Figure 5: Again, the concentration of IVT4 was not given. Figure 5a: is there transcriptional data? Does that mean that the induction of cell death in the combination requires RIG-I signaling while induction of IFN related genes are independent as suggested in Figure 4c? In Fig. 5b and 5c "cell death" (measured by CTG assay) or "cell viability" (FACS) of different treatments (Fig. 5b/ Fig5c) was compared. Why did the authors switch from the MTT based system as in Figure 4f? For sure, using different methods for the same experimental setting is more convincing, however, the terms "cell viability" and "cell death" used in this section are a bit confusing. Please explain why the viability of cell treated with IVT4 is inconsistent for different experiments: e.g. A549 cells, ~0.58 (16 hours treatment, Fig. 5f), ~0.3 (24 hours treatment, Fig. 6b) and ~0.75 (24 hours treatment, Fig. 6c). Fig. 5d: the comparison between IVT + IRF1o/e and o/e alone is not significant. Therefore the effect observed may only be additive. Also, it would be very interesting to see the effect of IRF1 k.o. in "osi+ IVT4" setting. Figure 5e-h: Please add a schematic explaining the tumor model (tumor induction, timing of IVT / inhibitor administration etc). As it stands, it is hard to follow. Why did the authors use distinct statistical tests in Figure 5e and 5f? What are the survival data of these mice? Please also provide information about the numbers of animals used in each group for Fig. 5f / 5g. Is that pooled data of independent experiments? In addition to the induction of IFN-target genes, a phenotypic characterization of the TME as in Figure 1f (T cells or NK cells, via FACS or IHC) during combination treatment in this humanized system would also be helpful to understand this story.

"The disjunction of the ATSC/EMT from the IFN gene expression profiles after osimertinib treatment ...and an absence of lymphocyte signaling in osimertinib/IVT-GAC treated tumors (Fig. S6g)." The sentence is too long and confusing, should be rephased.

Figure 6: This is a transplantable lung cancer model? Please briefly explain. Again, please provide a schematic, the numbers of animals used etc. What is Figure 6a and 6b? Tumors isolated from the mice on day XXX? Please provide survival data.

Discussion:

Page 21, line 24: The reference citation (4,48) was misplaced.

Page 22, line 9: "Considering the time dynamics, the lack of robust interferon secretion by treated cells, the limited engagement of the TBK1-IRF3 axis and the sustained activation following MAVS knock-out indicates a tumor cell-autonomous process distinct from previously described mechanisms." In light of their data presented in Figure 4f and 5a, the authors should discuss alternative mechanisms in more detail. Is cell death different from IFN gene induction? Does BRAFmut confer resistance to RIG-I agonists induced cell death?

Page 22, line 11: "sustained activation following MAVS knock-out" Is this description right (Fig. 4b)? Moreover, in the "previously described mechanisms", the experimental setting is similar, several cell lines are the same and both paper focus on EGFR inhibition. Furthermore, inhibiting mutant EGFR triggers Type I IFN-I upregulation via a RIG-I-TBK1-IRF3 pathway in the cited study. Thus, the differences between these two studies should be further discussed.

Minor comment:

In general, please add the oncogenic drivers of the cell lines used in the figures next to the graphs. Makes it easier to keep overview. Otherwise very confusing.

The font and font size of labeling in the figure should be the same, or at least comparable, this applies for all the figures and supplementary figures. And the labeling for the same event should be consistent, such as Fig1. b and c/d, x-axis labeling.

Reviewer #2:

Remarks to the Author:

The manuscript entitled "MAPK-mediated inflammatory reprogramming sensitizes tumors to targeted 2 activation of innate immunity sensor RIG-I" by Brägelmann et al. studies the causes of treatment-induced senescence in an effort to undermine the mechanisms by which cancers escape treatment. Beginning with transcriptomic analyses, the authors studied oncogene-driven cancer cells following treatment with driver-specific kinase inhibitors. The authors report a converse relationship between proliferative gene signatures (e.g., cMyc, E2F1, and G2/M signatures) and innate immune gene signatures (IFNa, IFNg_ and adult tissue stem cell (ATSC) gene signatures, along with a subset of known senescence-associated genes. The authors suggest that signaling through MAPK is sufficient to push this inverse transcriptomic balance towards increased proliferation with decreased innate immunity, while MAPK inhibition (using either a MPAK inhibitor or an inhibitor that blocks an upstream activator of MAPK, like EGFR or HER2 or MET) is sufficient to tip the balance towards greater innate immunity and less proliferation. The authors suggest that inhibition of proliferation or induction of cell death is not required for innate immune activation, however. Further, the authors show that agonist-inducible RIG-I signaling can be used to amplify the innate immune activation in conjunction with EGFR/MAPK inhibition to improve tumor growth inhibition.

The manuscript represents a good deal of data that is of high quality. The major attributes of the manuscript are the strong data showing combined efficacy of EGFR inhibition + RIG-I activation in lung tumors, which is a timely advancement for a very pressing and unmet need. The major criticism of the manuscript is the over-interpretation and mis-interpretation of the data in many places (noted below). It is possible that the manuscript might be better served if the data regarding Her2 inhibition in breast cancers was eliminated, as these data seem to muddy the interpretation considerably. Additional comments and suggestions are listed below. I would encourage the authors to continue along this line of very interesting and important work, and would be happy to see a resubmission upon addressing the reviewer comments.

Figure 1a and S1B. In all kinase inhibitor cell lines, transcriptomic evidence that cell cycle progression (genes associated with G2/M progression, E2F activity, and genes associated with cMyc) were down-regulated upon 3 days of kinase inhibitor treatment, while genes associated with innate immunity (IFNa, IFNg) and adult tissue stemness were up-regulated. Q: were there other gene groups that were altered, and these were the ones reported, or were these the only groups that were significantly altered?

FigS1A . Please rearrange the X-axis in FigS1A so that the treatment groups are more obvious.

Fig.S1B: The authors show evidence that the ATSC gene signature is elevated in response to treatment with inhibitors of oncogenic drivers (i.e., EGFR inhibitor in EGFR-driven lung cancer cells; ALK inhibitor in ALK-driven lung cancer cells; BRAF inhibitor in BRAF-mutant melanoma/colon cancer cells; pan-HER inhibitor in HER2-driven breast cancer cells). While the authors claim that the IFNa and IFNg gene signatures are also elevated in response to oncogenic inhibitors, it appears from the data that significant upregulation of IFNa/IFNg gene signatures occurs in only two groups (EGFR-driven lung cancers treated with EGFR inhibitor; BRAF mutant cancers treated with BRAF inhibitor), but not in three other groups (ALK-driven and MET-driven lung cancers treated with ALK inhibitor or MET inhibitor, respectively; HER2-amplified breast cancers treated with pan-HER inhibitor). The authors should make this more clear in their description of the data and in their interpretation.

The discrepancy in response between the EGFR inhibitor-treated lung cancer cells (which show upregulation of ATSC and IFNa/IFNg gene signatures) versus pan-HER inhibitor-treated breast cancer cells (which show upregulation primarily of ATSC, and little to no upregulation of IFNa/IFNg

gene signatures) may require additional study, since EGFR is inhibited by drug in both cases. Does the discrepancy come from the additional inhibition of other HER family members? Or does the discrepancy come from the fact that HER2 amplification drives signaling pathways that are diverse from EGFR-driven pathways? Or because one cancer type is breast, and one cancer type is lung? Fig1b-d and SuppFig1C-I. These experiments show data only for those cells treated with EGFR inhibitor. For consistency in interpretation, it is important to show that growth is blocked and senescence is induced in cells treated with the ALK inhibitor, the MET inhibitor, the BRAF inhibitor, and the pan-HER2 inhibitor.

Figure 1h. This figure is potentially very interesting, showing that innate immune gene signatures are elevated in HER2-amplified breast cancers following neoadjuvant anti-HER2 therapy. However, it is extremely important to note that neoadjuvant HER2 therapy is most often the monoclonal antibody trastuzumab (Herceptin), which would inherently alter the immune microenvironment in treated tumors due to antibody-mediated tumor cell cytotoxicity. This will need to be clarified, and the interpretation will need to be altered if indeed the tumors were treated with trastuzumab.

Figure 1K. It is unclear what type of tumor is assessed here and what the treatment conditions were.

FigS3D. It is unclear what this measures, and the scale of the bars seems exceedingly small compared to the entirety of the Y-axis.

Fig.S3f. The authors state that PI3K inhibition in PC9 cells failed to upregulate IRF1. However, the blot shown clearly shows upregulation of IRF1. This statement should be revised. Also, I think additional cell lines should be tested for response to PI3K inhibition.

Fig.S3H. Given that CD274 (PD-L1) is an IRF-activated gene and a key type I IFN-response gene, it is strange that its expression is downregulated in kinase inhibitor-treated cells, while most other IFN-response genes are upregulated.

Figure 3A. This is an interesting and important experiment, which attempts to separate the effects of inhibitor-induced cell death from inhibitor-induced signaling disruptions that lead to altered innate immune response in the treated cells. However, given that the authors used 10 μ M of a pan-caspase inhibitor that has an IC₅₀ of 400 nM for even the least sensitive caspases, and given that caspase-1 is rather important for sustaining an innate immune response, it will be important for the authors to repeat these experiments using an appropriate dose of a caspase-3/7 selective inhibitor.

Figure 5B: The relative viability for cells treated with tremetinib or vemurafinib should not be set at 100%, they should be set relative to the viable cells in the vehicle-treated group, as should the viable cells treated with the combination of vemurafinib/tremetinib plus IVT4, and cells treated with IVT4 alone.

Reviewer #3:

Remarks to the Author:

In this manuscript the authors performed transcriptional profiling of several oncogene-driven cancer cell lines after targeted therapy treatment and identified up-regulation of IFN target genes, as well as genes associated with therapy-induced senescence in conjunction with downregulation of genes associated with cell cycle progression. Using various model systems including patient-derived xenografts, humanized mouse models, syngeneic immunocompetent models, as well as analysis of patient samples and existing datasets, they identify MAPK inhibition-induced IRF1 upregulation as the primary mechanisms underlying increased expression of inflammatory genes, among which is RIG-1. They go on to show that treatment of EGFR-mutant lung cancer cells, and humanized xenografts with osimertinib followed by a RIG-1 agonist results in increased tumor cell death and decreased xenograft growth. Finally, in a syngeneic immunocomponent model of EGFR-mutant lung cancer, they show that the effects of RIG-1 agonist treatment are at least partially CD8+ T-cell and NK-cell dependent.

Overall, this study is of high scientific interest and of potential clinical importance. However, the following points should be addressed.

Major points:

1. The authors switch between model systems and disease models from figure to figure and this can make the manuscript difficult to follow. A model figure would be helpful to summarize their

results.

2. The authors initially claim that the effects of tumor cell up regulation of IFN are tumor cell autonomous, as it is not secreted into the culture media. However they also show in humanized mouse models that osimertinib treatment is associated with increased CD8+ T-cell infiltration (Fig 1f) and later show that the anti-tumor effects of a RIG-1 agonist are at least in part dependent on CD8+ T-cells and NK cells (Fig. 6e). Overall the impact of IFN-induction and RIG-1 agonist treatment on the tumor immune microenvironment remains incompletely characterized. For example, the mechanism underlying increased T-cell infiltration and how this may be affected by RIG-1 agonist treatment is not explored, nor is the affect of osimertinib or RIG-1 agonist treatment on innate immune cells within the tumor microenvironment.

Minor points:

1. The authors cite other studies showing that MAPK signaling can repress IFN gene expression, however, the mechanism underlying this effect is unclear. Experiments to assess this should be considered.

2. In figure 6 the authors utilize a syngeneic flank-model of EGFR-mutant lung cancer to assess the role of CD8 T-cells and NK cells in their system. An orthotopic model of lung cancer would provide a more relevant assessment of the role of the native immune-microenvironment in mediating the effects of a RIG-1 agonist.

REVIEWER COMMENTS

Reviewer #1 (Remarks to the Author):

MAPK-mediated inflammatory reprogramming sensitizes tumors to targeted activation of innate immunity sensor RIG-I
By Brägelmann et al.

In the manuscript, the authors uncovered that targeted inhibition of oncogenic signaling in different tumor types induced stemness- and senescence-associated reprogramming, featured as MAPK/IRF1-mediated inflammatory response in tumors. RIG-I agonist IVT4 can be used to maximize the anti-tumor efficacy of targeted therapy. The *in vitro* results were translated into mice models and achieved solid effect. The experiments are well designed and most of data are reliable. Furthermore, the idea of combining kinase inhibitors with RIG-I agonists against solid cancers is tempting. Nonetheless, substantial revision of the manuscript is required to support their claims.

We thank the reviewer for appreciating our work.

Introduction

The description regarding target therapy and drug resistance in the first part of discussion should be included in the introduction section. The effect of tumor-intrinsic activation of innate immunity networks should be briefly introduced, not only induction of cell death and immune recognition, but also their role in promoting immunotherapies (doi: 10.1126/sciimmunol.aau8943; doi: 10.1172/JCI131572, doi: 10.1016/j.celrep.2020.108080). This brings their findings with kinase inhibitors and RIG-I agonists in the relevant clinical context.

We thank the reviewer for this suggestion. We expanded the introduction regarding targeted therapy and drug resistance and include the mentioned references (please see page 5, lines 87-91).

Results

Figure 1:

In Fig. 1b and Fig. S1F, why no immunoblot band can be seen for p21? All three tested cells are p21 negative?

This point is well taken. Indeed, we were unable to detect relevant levels of p21 in all three EGFR cell lines after EGFR inhibition. To mitigate the possibility of technical artifacts precluding p21 detection we repeated the western blot including A549 cells as positive control since p21 induction following etoposide treatment has been described previously (Hoeflerlin et al., Genes Can 2011). A549 were treated with etoposide (72h, 50 μ M) and EGFR-mutant cell lines HCC4006, PC9 and HCC827 with osimertinib for 5d as in the original western blot. Actin serves as the loading control.

What does “TKI” stands for? Osimertinib? Same labeling should be used. What is the the oncogenic driver for PC9?

We are sorry for the confusion. TKI stands for Tyrosine Kinase Inhibitor, the oncogenic driver for PC9 is mutant EGFR. We clarified these points throughout the manuscript.

In Fig. S1H, legends for the color coding are missing.

We added the legend for color coding in Fig. S1H.

“higher levels of ... histone marks H3K9me3 and H3K27me3” is not well supported by Fig. S1I, since the loading control is not even, a quantitative immunoblot figure is necessary.

We have performed quantification of the western blots and added an appropriate figure (new Fig. S1i).

In Fig. S1I and J, is there a reason that proteins from different drug treatment time points were used for immunoblotting?

We do not observe major differences between the time points and thus the treatment period is of minor importance for this point. We have now repeated blots for experiments in Figure S1j with the same time points as in Fig. S1i.

For the humanized mouse model, Fig. 1f clearly showed T cells infiltration in the tumor treated with Osimertinib, however, no tumor volume change under Osimertinib treatment is shown.

This point is well taken. We included tumor volumes for these mice in the revised version of the manuscript (please see Figure S2g).

Thus, we have no idea if those infiltrating T cells are functional. What are the phenotypes of these T cells?

This is an important point. To address this issue we performed additional experiments including Hyperion imaging, which is able to not only validate tumor-infiltration with T cells, but also enables to visualize co-localization of granzyme staining in CD8 cells adjacent to tumor cells, which indicates cytolytic activity and thus a functional phenotype. These data are now included as Fig. 1h and Fig. S3a-c.

New Fig. 1h: Cytokerating (CK, cyan) staining marks tumor cells, green CD8 cells and red Granzyme B expression. Overlay of green and red yields yellow signals showing concurrent staining for CD8 and Granzyme B. This data is compatible with a presence of cytolytically active CD8 cells.

In addition, we set up an independent cohort of humanized mice to repeat the initial experiment and performed FACS experiments after osimertinib vs. vehicle treatment. The FACS analyses showed that treatment with osimertinib reduced markers of exhaustion in CD8 cells such as TIM3 and PD1, which further suggests a phenotypic change in lymphocytes following treatment. This data is presented in new Fig. 1i and discussed accordingly (please see page 8, lines 176-178 and page 22, lines 571-574).

In Fig. 1h, it would be better to show the figure for anti-HER2 base vs. post neoadjuvant therapy, if applicable.

Vice versa and to be consistent, Figure 1i and 1j should be divided in responders and non-responders, if applicable.

This point is well taken. To address this point we performed baseline vs. post neoadjuvant therapy analyses in the breast cancer cohort. Overall, the trends within the responder and the non-responder group are in agreement with the initial analysis albeit at a lower magnitude. Considering the argument of Reviewer #2 that patients were treated with trastuzumab we moved the original analysis and the new analyses to Fig. S3d.

In the melanoma cohort all patients responded with two patients having stable disease and the showing tumor reduction, which makes a division as responders vs. non-responders difficult. To incorporate treatment response to the induction of immune infiltration we however assessed time to progression compared to the added influx of CD8 cells as estimated from the RNA-seq. This analysis showed a significant correlation between longer time to progression and increased CD8 cells compared to baseline sample in the melanoma patients where both were available.

Figure 2: Fig. 2b/S3a: Did the patients “on” respond? Please provide explanation.

The melanoma patients in the original publication (Kwong et al., JCO 2015) were selected based on a RECIST-based tumor response under BRAFi or BRAFi+MEKi. Accordingly, all “on-treatment” patients in the Figure showed stable disease or better.

Fig. 2C: what is the consequence of these findings? Did these mice respond better than the untreated controls? This is important to assign the clinical relevance of these findings.

We would like to clarify this point. We used ex vivo cultured cells treated with a MEK inhibitor for these experiments. The viability data was shown in Fig. S3b and is now part of Fig. S4 in the revised version of the manuscript. We have not tested response of the donor mice to trametinib ourselves. However, our aim at this point is to show the degree of conservation regarding upregulation of an inflammatory response following MAPK inhibition across disease types and oncogenic drivers.

In Fig. S3D, it is not reasonable to set the y-axis value as 500 (U/ml) without a positive control (e.g. IVTA4 treated cells), the detection method might not work in this experiment.

We agree that the axis is difficult to interpret in the way it was displayed. It is certainly difficult to judge in S3D without a positive control in the same graph even though Type I IFN levels following IVT4 treatment were already included in Fig. SB with observed concentrations >10,000 U/ml. However, to clarify it at this point we included the standard curve used to benchmark the experiment. This clearly shows that measured levels are well below the lowest threshold and outside the detectable range (now Fig. S4d). It should be noted that this is a highly sensitive assay so that IFN is practically not detectable in the medium supernatants.

In the figure legends, D was marked as C.

We thank the reviewer for spotting this labeling error and corrected it.

In Fig. S3 E and F, why are different loading controls used for immunoblotting of the same cell line?

Depending on the size of the investigated proteins and species of the primary antibodies and fluorescence labels of the secondary antibodies we used Hsp90 or actin as loading controls.

Fig. 2e showed that inhibitor treatment lead to upregulation of VTCN1 (B7-H4), which was previously linked with resistance to immune checkpoint blockade (ICB). In this context, it would be very interesting to demonstrate the phenotype of t infiltrated T cells demonstrated in Fig. 1f, to see the impact of inhibitor treatment on T differentiation and especially if they express an "exhausted" T cell phenotype.

We thank the reviewer for this suggestion. In order to further characterize the T cell phenotype we analyzed a new cohort of humanized mice and investigated the phenotype of the T cells. As discussed above we observe a lower rate of CD8 cells positive for TIM3 and PD1 following osimertinib treatment than after vehicle treatment. Similarly Hyperion imaging allowed us to investigate the functional phenotype of invading T cells (Fig. 1h, Fig. S3a-c). Interestingly, treatment with IVT4 further reduced exhaustion phenotype of CD8 cells in the humanized mouse model. This data is presented in Fig. 1i and discussed on page 22 (lines 571-574).

What about PD-L1 expression in the tumor cells treated with those inhibitors in Figure 2e, given that type I interferon signaling induces PD-L1 expression.

We thank the reviewer for this suggestion. It is indeed surprising that we observe both induction of interferon target genes and mRNA downregulation of PD-L1 in those cell lines. To validate these findings we performed FACS analyses of the cell lines after osimertinib and trametinib treatment in vitro (new Fig. S4n).

Similarly, we also analyzed CD45 negative (i.e. the non-hematological) cells from tumors in the new cohort of humanized PC9 xenografts. Here we observed a similar decrease of PD-L1+ cells (new Fig. S4o).

Overall, while indeed IFN triggers PD-L1 expression a recent report provided evidence that EGFR may directly contribute to increased PD-L1 levels by stabilizing PD-L1 mRNA (Ghosh et al., *Cell Rep* 2021).

The introduction of DUSP6 and SPRY4 as “two negative feedback regulators of MAPK signaling” somewhat detracts from the main story and causes confusion. I suggest to move the DUSP6 data into supplement. Moreover, why is the x-axis labeling in Fig 2h and 2i different? And also, for Fig. S3N, x-axis and y-axis labelings are different from others. Again, do these results reflect the patient outcome?

We are sorry if these panels cause confusion. We showed data for DUSP6 and SPRY4 as representative genes that are inversely correlated with MAPK activity status. A broader overview about MAPK activity is given in the boxplots combining genes taken from a recently proposed MAPK pathway score (Wagle et al., *Nat Prec Onc* 2018). The data sources were different for Fig. 2h and 2i/S3N and the labeling of the x-axes was therefore not identical. We agree that a unified labeling is beneficial and adjusted it to improve clarity in Fig. 2h,l and Fig. S4q-t.

Regarding patient outcome we performed survival analyses for proportion of CD8 T cells (left, Log-rank p=0.83) and DUSP6 expression (right, Log-rank p=0.17) in the TCGA LUAD cohort for which cell type proportions were successfully estimated by CIBERSORT (n=350 patients).

Grouping patients as high/low based on the median of the respective parameter did not indicate differences in survival. However, the goal of our analysis at this point was to investigate whether a potential association between MAPK activation and infiltration with CD8 cells. Based on the in vitro and in vivo observations that decreased MAPK activity induced by inhibitor treatment increased inflammatory signaling and influx of T cells we aimed to assess the relationship in patient samples. Observing the inverse relationship between MAPK activity (as indicated by high levels of DUSP6, SPRY4 or MAPK scores) and decreased proportions of CD8 cells supports this hypothesis. It would be nice to also observe differences in survival. However, the lack thereof may also indicate the a two-hit strategy with a kinase inhibitor leading to altered immune infiltration followed by a RIG-I agonist may still benefit patients.

Figure 3: “This data indicates that inhibition of the MAPK pathway and not indirect cell death signaling ...” What does “indirect cell death signaling” mean here?

We are sorry about not being clear enough at this point. By “indirect cell death signaling” we meant that the inflammatory effects are independent of the occurrence of cell death. It has been shown that cell death can induce an inflammatory response (e.g. Zitvogel et al., *Cell* 2010). Using Caspase-inhibition to prevent cell death we thus aimed to segregate cell intrinsic from cell death related reactions. We changed this passage accordingly to clarify this point.

In Fig. 3h, the authors showed that IRF1 knock-out significantly reduced transcriptional induction of several IFN target genes. What about the effect of IRF-1 knockout on cell death induced by the inhibitors?

We thank the reviewer for this suggestion. We tested the viability in response to osimertinib in IRF1 knock-out cells to osimertinib (GI_{50} 30nM, left) and observed no relevant difference compared to empty control cells (GI_{50} 80nM, right).

In agreement with viability data, we observed comparable rates of apoptosis induction following osimertinib mono treatment in e.v. cells (Fig. 5a) and IRF1 knock-out cells (new Fig. S7c). Overall sensitivity to osimertinib does not appear to be changed by IRF1 knock-out.

Figure 4: The concentration of IVT4 used in all the experiments of Figure 5 was not provided in the figure or figure legend.

We added the IVT4 concentration in the respective legends.

Figure 4b: is there a transcriptional profile as in c and cell viability/death data?

We thank the reviewer for this suggestion. To further assess the impact of IVT4 on the cellular signaling we have now performed RNA-seq of PC9 cells and A549 cells. We observe massive induction of inflammatory and interferon signaling as well as of apoptosis genes including NOXA and PUMA (new Fig. 4e,f).

	Geneset	NES	FDR q-val
1	Hallmark Interferon Gamma Response	10.39	0.000
2	Hallmark Interferon Alpha Response	9.61	0.000
3	Hallmark TNFA Signaling via NFKB	7.86	0.000
4	Hallmark Inflammatory Response	5.16	0.000
5	Hallmark Allograft Rejection	3.60	0.000
6	Hallmark IL6 JAK STAT3 Signaling	3.49	0.000
7	Hallmark Apoptosis	3.25	0.000
8	Hallmark Complement	3.23	0.000
9	Hallmark IL2 STAT5 Signaling	2.88	0.000
10	Hallmark KRAS Signaling UP	2.82	0.000

In addition, we validated the effects of osimertinib, IVT4 and osi+IVT4 in PC9 e.v., sgMAVS or sgSTING cells by qPCR for several inflammatory genes (new Fig. S7a).

Cell death data for the PC9 e.v and sgMAVS can be seen in Fig. 5a.

“While TLR expression was generally low in the cell lines, response to MDA5 and cGAS agonists was limited despite measurable RNA levels (Fig. S5d).” Wrong supplementary figure was cited here.

We corrected this labeling error accordingly.

Page 15. Line 24: at this point, it should be stated that these findings are in line with a recent report by Heidegger et al, (doi: 10.1126/sciimmunol.aau8943) where RIG-I agonist seemed more potent than cGAS/STING agonists.

We thank the reviewer for pointing this out and included the reference at this position.

For STING-knockout PC9 cells, an immunoblot figure is required to confirm the knockout result.

We would like to point out that STING is essentially non-detectable on a protein level in parental PC9 cells (Fig. S6d, top), which is in line with the lack of response to STING pathway agonists (Fig 4a). For this reason, we turned to an NGS-based strategy to verify the knock-out, but in addition also included an immunoblot of STING with HCC827 as a positive control for STING protein (new Fig. S6d, bottom).

In Fig. 4f, the relative viability of PC9 cell decreased to 0.6 after 16 hours IVT4 treatment, thus, an additional figure of longtime treatment (up to three days) would be interesting.

We thank the reviewer for this suggestion. We performed longer-term viability assays and show an increasing effect over time for all cell lines investigated. We included these experiments in the revised version of the manuscript (please see Fig. S6j).

Furthermore, why do some of the oncogene-driven cell lines not respond (e.g. A375, Colo205) to IVT4? Are these the BRAFmut cell lines as in in figure 5 and if yes, does that interfere with RIG-I signaling?

We agree with the reviewer that the variability in response to IVT4 after 16h in Fig. 4f (new Fig. 4g) is intriguing. Indeed, A375 and Colo205 are BRAFmut cell lines also used in Fig. 5. Based on this reviewer's suggestion we performed viability assays up to 72h with a panel of cell lines including A375 and Colo205 and observed decreased viability for all cell lines. Even though the effect of IVT4 mono treatment is limited after 16h, prolonged exposure thus appears to impact BRAF mutant cells as well. Whether this is specific to the two cell lines or to BRAF mutant cells in general is difficult to judge from the present data and the available literature.

Figure 5: Again, the concentration of IVT4 was not given.

We added the concentration in the figure legend.

Figure 5a: is there transcriptional data?

We thank the reviewer for this suggestion. To assess the transcriptional response to IVT4 in PC9 cells we generated new RNA-seq (new Fig. 4e, f). In addition we analyzed a number of key inflammatory genes by qPCR to study the effects of IVT4, osimertinib and a combination of both. The data is now included as Fig. S7a.

Does that mean that the induction of cell death in the combination requires RIG-I signaling while induction of IFN related genes are independent as suggested in Figure 4c?

Indeed we hypothesize that the induction of IFN target genes by osimertinib is independent of RIG-I/MAVS signaling as shown in Fig. 4c, while the enhanced induction of cell death with IVT4 requires a functional RIG-I/MAVS axis.

In Fig. 5b and 5c “cell death” (measured by CTG assay) or “cell viability” (FACS) of different treatments (Fig. 5b/ Fig5c) was compared. Why did the authors switch from the MTT based system as in Figure 4f? For sure, using different methods for the same experimental setting is more convincing, however, the terms “cell viability” and “cell death” used in this section are a bit confusing.

We are sorry for the lack of clarity. In the mentioned figure panels we estimated the effects on cell death by using AnnV/PI FACS to complement our investigations of effects on cell viability. To assess cell viability we used either CTG or MTT, which both measure metabolic activity albeit with slightly different methodological approaches. The reason for using both approaches was based on experiments being performed in different laboratories with one lab using MTT and the other one performing CTG assays. We agree that the use of both approaches may seem confusing at first sight. However, overall we consider it a strength of our study that we observe similar results using different methods across various laboratories, which indicates the robustness of our findings. To minimize confusion we aimed to adjust the labeling in order to make this clearer.

Please explain why the viability of cell treated with IVT4 is inconsistent for different experiments: e.g. A549 cells, ~0.58 (16 hours treatment, Fig. 4f), ~0.3 (24 hours treatment, Fig. 5b) and ~0.75 (24 hours treatment, Fig. 5c).

This point is well taken. The variability in results for A549 in the mentioned figures are most likely explained by different experimental approaches. In Fig. 4f (new Fig. 4g) we used an MTT assay after 16h treatment. In contrast, for Fig. 5b cells were grown for 2d in DMSO to provide a control to trametinib treated cells prior to treatment with IVT4 for 24h before viability was assessed with CTG in 96-well plates. For experiments in Fig. 5c we had a comparable set-up as in 5b, but in 6-well plates. However, we measured induction of cell death by AnnV/PI FACS rather than metabolic activity as a proxy for cell viability. In addition, variability due to different batches of synthesized IVT4, in IVT4 transfection efficiency etc. are conceivable, but most likely not of major impact to the systematic differences in experimental designs.

Fig. 5d: the comparison between IVT + IRF1o/e and o/e alone is not significant. Therefore the effect observed may only be additive.

We thank the reviewer for this comment. Indeed, we had initially assessed the differences between e.v. and IRF1oe with IVT-GAC and between e.v. and IRF1oe with IVT4 by t-tests. A statistical test between IVT4+IRF1oe and IVT-GAC+IRF1oe had not been performed. According to the comment we have now re-assessed our data and performed a two-way ANOVA incorporating IRF1oe/IVT4 treatment, adjusting for biological replicate followed by Tukey post-hoc test for multiple comparisons. Now the statistical test of IRF1oe +/- IVT4 is also included. The updated p-values are now included in the revised Fig. 5d.

Similarly, we observe a significant interaction for IVT4 in IRF1oe vs e.v. when performing the ANOVA including an term modeling statistical interaction of IVT4_treatment with oe_type, followed by Tukey post-hoc analysis. While heterogeneity exists between replicates due to

transient and repeated transfection with o/e constructs followed by IVT4/IVT-GAC, this further indicates that IRF1oe contributes to IVT4 sensitivity in a multiplicative rather than additive fashion.

Also, it would be very interesting to see the effect of IRF1 k.o. in “osi+ IVT4” setting.

We thank the reviewer for the suggestion. We performed the required experiments and noticed that in the context of IRF1 k.o. both IVT4 and osi+IVT4 still induced relevant levels of cell death as measured by Annexin V/PI staining (new Fig. 7c).

IRF1 knock-out does not completely abrogate RIG-I induction by osimertinib and other IRF-family members like IRF7 and IRF9 that may compensate for IRF1 loss are still induced following osimertinib treatment (new Fig. S5i):

Nevertheless, we wanted to further understand the mechanism of and contributors to cell death induced by IVT4+/- osimertinib. We therefore initiated experiments involving cell death inhibitors preventing necroptosis (Nec1s), ferroptosis (Fer1) and caspase-dependent cell death (ZVAD). These experiments showed that IVT4 induced caspase-dependent cell death as measured by PI staining without evidence of necroptosis or ferroptosis (new Fig. S7d).

In our initial manuscript we had noticed an induction of anti-apoptotic BCL2 family members in PC9 and HCC827 cells under treatment (Fig. S1j). Similarly, others have described a higher susceptibility towards BCL2 or MCL1-inhibitors in cells treated with kinase inhibitors (e.g. Hata et al., Nat Med 2016). In our RNA-seq experiments we observed enrichment of an apoptotic gene signature including the pro-apoptotic proteins NOXA and PUMA after IVT4 treatment. To assess whether NOXA and PUMA may be involved in IVT4-induced cell death we performed knock-down experiments. Indeed, knock-down of NOXA significantly reduced cell death induction following osi+IVT4 in PC9 cells, while PUMA knock-down did not contribute significantly (new Fig. S7e,f).

Taken together the data collected during the revisions indicates that the synergistic effect of combining osi + IVT4 may depend on two factors: On one hand, tumors cells express higher levels of RIG-I with osimertinib, which may enhance response to IVT4. Simultaneously, surviving cells become more susceptible to induction of apoptosis and more reliant on anti-apoptotic BH3 family members in order to survive. Upregulation of NOXA by IVT4 may thereby further enhance cell killing. At the same time IVT4 induces a strong inflammatory response that is independent of the INF response signature of kinase inhibitors that may be of relevance for the overall therapeutic response for combination therapies.

Figure 5e-h: Please add a schematic explaining the tumor model (tumor induction, timing of IVT / inhibitor administration etc). As it stands, it is hard to follow. Why did the authors use distinct statistical tests in Figure 5e and 5f? What are the survival data of these mice? Please also provide information about the numbers of animals used in each group for Fig. 5f / 5g. Is that pooled data of independent experiments?

We would like to apologize for the lack of clarity. According to the suggestion we added a schematic figure for the experiment original 5e and the animal numbers. It should be noted that original Fig. 5e (now Fig. S8b) is a mouse xenografts of KRAS-mutant A549 lung cancer cells treated with trametinib or vehicle + IVT4 or IVT-GAC, which were given concurrently. A schematic can be found in Fig. S8a. In contrast, original Figs. 5f/g (now Fig. 5e,g) show data of

humanized EGFR-mutant PC9 xenografts that were treated with osimertinib for a lead-in period before IVT4 was administered. We also included a new schematic in Fig. 5e. Since humanized mice eventually develop graft-vs-host disease treatment time is limited. For this reason osimertinib was stopped with the onset of IVT4 treatment in order to observe IVT4 effects unmitigated by ongoing osimertinib response. For longer term treatment we also used the syngeneic mouse model and as an additional control also performed a similar set-up in a non-humanized PC9 xenograft (new Fig. S8f).

The data was obtained from several mice that were implanted with one tumor per flank and thus carried one or two tumors based on engraftment rate. Each experiment in the presented figures was performed once, however we have since validated the higher efficacy on tumor shrinkage of osimertinib + IVT4 in an additional cohort of humanized mice (new Fig. S8d). Due to the fact that mice carried up to two tumors we performed an analysis adjusting per mouse for original Fig. 5f. Following the valuable comment regarding different statistical tests between Fig. 5e and 5f (now Fig. S8b and 5e, respectively) we now also compared the effect of IVT4 in the trametinib or vehicle treated groups using an ANOVA adjusting per mouse to avoid mouse-specific effects. The significance and conclusions remain unchanged.

In addition to the induction of IFN-target genes, a phenotypic characterization of the TME as in Figure 1f (T cells or NK cells, via FACS or IHC)) during combination treatment in this humanized system would also be helpful to understand this story.

We thank the reviewer for this suggestion. We repeated the experiment and performed FACS analyses of infiltrating cells in the humanized xenografts model after 6d of IVT4 treatment. Interestingly, we observed a lower proportion of exhausted CD8 cells as indicated by lower proportions of TIM3+,PD1+,CD8+ cells in osi IVT4 treated compared to osi IVT-GAC treated animals (new Fig. 5f). This difference was neither present in CD4 cells nor did we observe such differences in veh IVT4 vs. veh IVT-GAC tumors (new Fig. 8d). This indicates that IVT4 may contribute to a more active cytotoxic T cell response.

Moreover, we performed an additional cohort of mice in the syngeneic mouse model treated with osi or osi+IVT4. Interestingly, we did not observe differences in the levels of infiltrating T or NK cells with or without IVT4. However, expression of PD1 was significantly lower in CD8 cells in the osi+IVT4 arm (new Fig. 6e and 9e). This indicates that IVT4 may help to induce a less exhausted CD8 phenotype in this setting as well.

“The disjunction of the ATSC/EMT from the IFN gene expression profiles after osimertinib treatment ...and an absence of lymphocyte signaling in osimertinib/IVT-GAC treated tumors (Fig. S6g).” The sentence is too long and confusing, should be rephased.

We rephrased the sentence accordingly (please see page 18, lines 456-459).

Figure 6: This is a transplantable lung cancer model? Please briefly explain. Again, please provide a schematic, the numbers of animals used etc. What is Figure 6a and 6b? Tumors isolated from the mice on day XXX? Please provide survival data.

This point is well taken. To investigate the effects of EGFR inhibition on inflammatory signaling in tumor cells and the TME in a fully immunocompetent environment we used the syngeneic EGFR-mutant mouse model described in Ohashi et al. (Can Sci 2008). Established lung tumors from C57BL/6 mice are transplanted into the flanks of C57BL/6 mice. Compared to existing autochthonous EGFR-mutant lung cancer mouse models e.g. Politi et al. (Genes Dev, 2006) this model has the advantage that it does not show a complete and very durable response to EGFR inhibition, but develops resistance after prolonged treatment (Higo et al., Lung Cancer 2019). It thereby reflects the clinical situation well. Moreover, it has the advantage to allow IVT4 administration via intratumoral injection, which facilitates the proof of concept of our proposed treatment schedule. Other modes of administration such as inhalation have been used e.g. with TLR agonists, but not yet for IVT4 and thus represent a promising avenue for future research.

Fig. 6a and 6b show RNA-seq of mice treated with osimertinib. The corresponding tumor volume data is shown in Suppl. Fig. S9a with the legend indicating the number of animals and tumors. We moved this information to the legend of the main figure for clarification. As in previous animal experiments this study was not geared for survival differences due to the long duration until EGFR resistance develops. Therefore mice were sacrificed at the indicated time points/at the end of the study period.

Discussion:

Page 21, line 24: The reference citation (4,48) was misplaced.

We corrected the reference citation.

Page 22, line 9: “Considering the time dynamics, the lack of robust interferon secretion by treated cells, the limited engagement of the TBK1-IRF3 axis and the sustained activation following MAVS knock-out indicates a tumor cell-autonomous process distinct from previously described mechanisms.” In light of their data presented in Figure 4f and 5a, the authors should discuss alternative mechanisms in more detail. Is cell death different from IFN gene induction? Does BRAFmut confer resistance to RIG-I agonists induced cell death?

We thank the reviewer for this remark. We have expanded the discussion based on this comment. Overall, we propose that the induction of an IFN response following inhibition of an

oncogenic driver such as EGFR in the studied models is due to a cell-autonomous mechanism. Our data suggests that cell death is not required for this response as indicated by the experiments with a pan-caspase inhibitor (Fig. 3a) or the more Caspase 3 specific inhibitor DEVD (new S5a) performed during the revisions.

Inflammatory signaling thus appears to be independent of cell death. Moreover, we have now also performed an IFN receptor (*IFNAR1*) knock-out in PC9 cells (new Fig. S4e). Since we were unable to obtain a signal for IFNAR1 by Immunoblot we used a functional assay by treating with recombinant IFN α (16h) to validate the knock-out:

While the knock-out severely decreased IFIT1 increase in response to recombinant IFN α treatment in three different knock-out clones (left) the response was not significantly different with osimertinib treatment. Interestingly, treatment with IFN α did not induce IRF1, while osimertinib did - irrespective of knock-out status (right). Together with the lack of detectable IFN levels by two different methods in medium supernatant treated of treated cells this renders autocrine/paracrine effects due to IFN secretion unlikely.

Considering our data on MAPK inhibition inducing a similar response we therefore consider repression of IRF1 by active MAPK signaling and a de-repression following MAPK inhibition a major mechanism in the observed phenotype. To further assess this point we included experiments using an ERK-inhibitor (SCH772984), which also induced IRF1 and IFIT1 (new S4j).

Moreover we used ChIP-qPCR analyses to study transcriptional rate as measured by RNA Pol II occupancy on IRF1, VTCN1 and DUSP6. As expected both IRF1 and VTCN1 showed significantly increased occupancy of total Pol II and of Serine 2 and 6 phosphorylated –and thus activated– Pol II at the transcription start site (TSS), in the gene body (GB) and at transcription termination (Term.). This indicates that transcriptional activity for both genes is enhanced following 72h osimertinib treatment (new Fig. 3k,l).

DUSP6, a major negative feedback regulator of MAPK signaling is significantly repressed and devoid of Pol II binding during osimertinib treatment (new Fig. S5j).

Overall, our data suggest that a de-repression of IRF1 and potentially other IRF family members may drive the inflammatory response observed upon MAPK pathway inhibition.

Page 22, line 11: “sustained activation following MAVS knock-out” Is this description right (Fig. 4b)? Moreover, in the “previously described mechanisms”, the experimental setting is similar, several cell lines are the same and both paper focus on EGFR inhibition. Furthermore, inhibiting mutant EGFR triggers Type I IFN-I upregulation via a RIG-I-TBK1-IRF3 pathway in the cited study. Thus, the differences between these two studies should be further discussed.

We thank the reviewer for pointing this out. We have now further expanded the discussion regarding the different results of the studies including the IFNAR1 knock out data. As detailed above we further investigated the effect of IFN α by performing *IFNAR1* knock-out in PC9 cells. Overall, sensitivity to osimertinib did not change relevantly in *IFNAR1* knock-out cells (left, GI₅₀ 40nM) compared to empty vector control cells (right, GI₅₀ 60nM).

More importantly, *IFNAR1* knock-out did not significantly impact induction of IFIT1 or IRF1 following IFN α treatment. Using IRF1 knock-out we can show that several IFN target-genes are induced to a significantly weaker degree following osimertinib treatment. At the same time IRF1 is not induced by treatment with recombinant IFN α in PC9 cells. Together with the very low/undetectable levels of IFN α in the supernatant of osimertinib treated cells secreted IFN does not appear to be a major driver in our models, while inhibition of the MAPK pathway seems to underlie the effects we describe. However, it should be noted that several differences exist between both studies, e.g. use of EGFR inhibitor (erlotinib vs. osimertinib), genetic manipulation (knock-downs vs. knock-outs), use of in vivo systems (PDX, xenografts vs. PDX, xenograft, humanized xenografts and syngeneic mouse model). Interestingly, Gong et al. observe no effect in EGFR-mutant HCC827 xenografts treated with erlotinib (Gong et al., Nat Can 2020, Fig. 3m; Fig. 7a).

Apart from methodological differences it is very well conceivable that different mechanisms may be involved in the interferon response or may potentially be involved at different points in time. We rephrased the discussion of this data in the light of the reviewer's comments (please see page 21, lines 537-548).

Minor comment:

In general, please add the oncogenic drivers of the cell lines used in the figures next to the graphs. Makes it easier to keep overview. Otherwise very confusing.

We thank the reviewer for this comment. We have now annotated the oncogenic driver where possible.

The font and font size of labeling in the figure should be the same, or at least comparable, this applies for all the figures and supplementary figures. And the labeling for the same event should be consistent, such as Fig1. b and c/d, x-axis labeling.

We thank the reviewer for this suggestion. We have accordingly further harmonized figure layout and labeling.

Reviewer #2 (Remarks to the Author):

The manuscript entitled "MAPK-mediated inflammatory reprogramming sensitizes tumors to targeted 2 activation of innate immunity sensor RIG-I" by Brägelmann et al. studies the causes of treatment-induced senescence in an effort to undermine the mechanisms by which cancers escape treatment. Beginning with transcriptomic analyses, the authors studied oncogene-driven cancer cells following treatment with driver-specific kinase inhibitors. The authors report a converse relationship between proliferative gene signatures (e.g., cMyc, E2F1, and G2/M signatures) and innate immune gene signatures (IFN α , IFN γ and adult tissue stem cell (ATSC) gene signatures, along with a subset of known senescence-associated genes. The authors suggest that signaling through MAPK is sufficient to

push this inverse transcriptomic balance towards increased proliferation with decreased innate immunity, while MAPK inhibition (using either a MPAK inhibitor or an inhibitor that blocks an upstream activator of MAPK, like EGFR or HER2 or MET) is sufficient to tip the balance towards greater innate immunity and less proliferation. The authors suggest that inhibition of proliferation or induction of cell death is not required for innate immune activation, however. Further, the authors show that agonist-inducible RIG-I signaling can be used to amplify the innate immune activation in conjunction with EGFR/MAPK inhibition to improve tumor growth inhibition.

The manuscript represents a good deal of data that is of high quality. The major attributes of the manuscript are the strong data showing combined efficacy of EGFR inhibition + RIG-I activation in lung tumors, which is a timely advancement for a very pressing and unmet need. The major criticism of the manuscript is the over-interpretation and mis-interpretation of the data in many places (noted below). It is possible that the manuscript might be better served if the data regarding Her2 inhibition in breast cancers was eliminated, as these data seem to muddy the interpretation considerably. Additional comments and suggestions are listed below. I would encourage the authors to continue along this line of very interesting and important work, and would be happy to see a resubmission upon addressing the reviewer comments.

We thank the reviewer for the overall appreciation of our findings and for the encouraging comments regarding the potential clinical relevance.

Figure 1a and S1B. In all kinase inhibitor cell lines, transcriptomic evidence that cell cycle progression (genes associated with G2/M progression, E2F activity, and genes associated with cMyc) were down-regulated upon 3 days of kinase inhibitor treatment, while genes associated with innate immunity (IFN α , IFN γ) and adult tissue stemness were up-regulated. Q: were there other gene groups that were altered, and these were the ones reported, or were these the only groups that were significantly altered?

This is a very relevant question. The gene sets displayed were those that appeared conserved across the panel of cell lines. However, other gene sets also became significant. For a comprehensive overview we have now added tables of the Hallmark gene sets for all cell lines as Supplementary Table S1.

FigS1A . Please rearrange the X-axis in FigS1A so that the treatment groups are more obvious.

We thank the reviewer for this suggestion and have rearranged Fig. S1A accordingly.

Fig.S1B: The authors show evidence that the ATSC gene signature is elevated in response to treatment with inhibitors of oncogenic drivers (i.e., EGFR inhibitor in EGFR-driven lung cancer cells; ALK inhibitor in ALK-driven lung cancer cells; BRAF inhibitor in BRAF-mutant melanoma/colon cancer cells; pan-HER inhibitor in HER2-driven breast cancer cells). While the authors claim that the IFN α and IFN γ gene signatures are also elevated in response to oncogenic inhibitors, it appears from the data that significant upregulation of IFN α /IFN γ gene signatures occurs in only two groups (EGFR-driven lung cancers treated with EGFR inhibitor; BRAF mutant cancers treated with BRAF inhibitor), but not in three other groups (ALK-driven and MET-driven lung cancers treated with ALK inhibitor or MET inhibitor, respectively; HER2-amplified breast cancers treated with pan-HER inhibitor). The authors should make this more clear in their description of the data and in their interpretation.

We thank the reviewer for this point. Indeed, response was stronger in BRAF-mut and EGFR-mut cells, however also present in Her2-mutant SKBR3 cells. In light of the reviewer's comment, we have carefully adjusted the interpretation and discussion of the results accordingly.

The discrepancy in response between the EGFR inhibitor-treated lung cancer cells (which show upregulation of ATSC and IFN α /IFN γ gene signatures) versus pan-HER inhibitor-treated breast cancer cells (which show upregulation primarily of ATSC, and little to no upregulation of IFN α /IFN γ gene signatures) may require additional study, since EGFR is inhibited by drug in both cases. Does the discrepancy come from the additional inhibition of other HER family members? Or does the discrepancy come from the fact that HER2 amplification drives signaling pathways that are diverse from EGFR-driven pathways? Or because one cancer type is breast, and one cancer type is lung?

We thank the reviewer for this question. To further clarify this issue we have performed immunoblots of lung cancer cells (PC9) and breast cancer cell lines (SKBR3 and BT474) after treatment with EGFR-inhibitor osimertinib or pan-Her inhibitor neratinib:

Based on these results it becomes clear that induction of IRF1 and IFIT1 in PC9 cells is present with both inhibitors. At the same time, SKBR3 and BT474 show induction of IRF1 with neratinib and –to a lower degree – following osimertinib. However, IFIT1 is not induced in neither of them. This indicates that certain pathways are conserved between all cell lines and that the induction of IRF1 is a shared principle across entities, but that specific signatures of the inflammatory response itself may be distinct between different tumor entities and/or oncogenic drivers.

Fig1b-d and SuppFig1C-I. These experiments show data only for those cells treated with EGFR inhibitor. For consistency in interpretation, it is important to show that growth is blocked and senescence is induced in cells treated with the ALK inhibitor, the MET inhibitor, the BRAF inhibitor, and the pan-HER2 inhibitor.

We thank the reviewer for this suggestion. We repeated the experiments with the indicated inhibitors.

We counted cells after 5d treatment (new Fig. S1l):

We performed cell cycle experiments after 5d (new Fig. S1m):

We also stained cells for beta-galactosidase (new Fig. 1k, left ctrl, right 5d treatment):

Despite the intrinsic heterogeneity of the cell lines panel the effects are comparable and therefore the collected data strongly support our initial findings.

Figure 1h. This figure is potentially very interesting, showing that innate immune gene signatures are elevated in HER2-amplified breast cancers following neoadjuvant anti-HER2 therapy. However, it is extremely important to note that neoadjuvant HER2 therapy is most often the monoclonal antibody trastuzumab (Herceptin), which would inherently alter the immune microenvironment in treated tumors due to antibody-mediated tumor cell cytotoxicity. This will need to be clarified, and the interpretation will need to be altered if indeed the tumors were treated with trastuzumab.

We thank the reviewer for this very valuable point. We obtained the RNA-seq data from a published study by Varadan and colleagues (Clin Can Res 2016), in which patients were indeed treated with trastuzumab. We added this information explicitly in the text and adjusted the interpretation accordingly. We also moved all figures regarding this cohort to the supplement in the light of the reviewer's comments.

Figure 1K. It is unclear what type of tumor is assessed here and what the treatment conditions were.

We are sorry if the labeling was not clear enough. The data is from melanoma patients also used in Fig. 1i and 1j. We adjusted the legend to make this clear.

FigS3D. It is unclear what this measures, and the scale of the bars seems exceedingly small compared to the entirety of the Y-axis.

We would like to apologize for the confusion. We adjusted the axis/added the standard curve to give a meaningful comparison (please see also comment of Reviewer 1 above).

Fig.S3f. The authors state that PI3K inhibition in PC9 cells failed to upregulate IRF1. However, the blot shown clearly shows upregulation of IRF1. This statement should be revised. Also, I think additional cell lines should be tested for response to PI3K inhibition.

We thank the reviewer for this suggestion. Indeed, a slight induction of IRF1 can be observed in PC9 cells with mono PI3K inhibition even though much less than with MEK-inhibition or combined MEK- and PI3K-inhibition. We have rephrased our interpretation of the Immunoblot results in light of these findings. Additionally we performed Immunoblot for a panel of additional cell lines (new Fig. S4h):

Overall, the response of IRF1 and IFIT1 is stronger with trametinib although partial induction can be observed with apitolisib in HCC4006 cells. We speculate that this heterogeneity might be linked with distinct feedback cross-regulation of both pathways and we present this point now in the light of the reviewer's comments (please see page 10, lines 241-243).

Fig.S3H. Given that CD274 (PD-L1) is an IRF-activated gene and a key type I IFN-response gene, it is strange that its expression is downregulated in kinase inhibitor-treated cells, while most other IFN-response genes are upregulated.

We agree with the reviewer that this is indeed puzzling. As mentioned above we have now further verified this downregulation on a protein level using FACS (new Fig. S4n).

Furthermore, we observed a similar decrease of PD-L1+ cells in vivo (new Fig. S4o).

Since IFN triggers PD-L1 expression a recent report provided evidence that EGFR may directly contribute to increased PD-L1 levels by stabilizing PD-L1 mRNA (Ghosh et al., *Cell Rep* 2021). It is therefore conceivable that inhibition of EGFR may destabilize PD-L1 mRNA and thereby offset the positive effects by the IFN-like response.

Figure 3A. This is an interesting and important experiment, which attempts to separate the effects of inhibitor-induced cell death from inhibitor-induced signaling disruptions that lead to altered innate immune response in the treated cells. However, given that the authors used 10 uM of a pan-caspase inhibitor that has an IC50 of 400 nM for even the least sensitive caspases, and given that caspase-1 is rather important for sustaining an innate immune response, it will be important for the authors to repeat these experiments using an appropriate dose of a caspase-3/7 selective inhibitor.

We thank the reviewer for appreciating our approach to deconvolute more direct signaling effects from those due to cell death, which may generate secondary effects. We have now repeated the qPCR analyses with caspase 3/7 inhibitor DEVD with overall comparable results (new Fig. S5a).

Figure 5B: The relative viability for cells treated with tremetinib or vemurafinib should not be set at 100%, they should be set relative to the viable cells in the vehicle-treated group, as should the viable cells treated with the combination of verafinib/tremetinib plus IVT4, and cells treated with IVT4 alone.

This point is well taken. We have changed the figure accordingly (new Fig. 5b):

Reviewer #3 (Remarks to the Author):

In this manuscript the authors performed transcriptional profiling of several oncogene-driven cancer cell lines after targeted therapy treatment and identified up-regulation of IFN target genes, as well as genes associated with therapy-induced senescence in conjunction with downregulation of genes associated with cell cycle progression. Using various model systems including patient-derived xenografts, humanized mouse models, syngeneic immunocompetent models, as well as analysis of patient samples and existing datasets, they identify MAPK inhibition-induced IRF1 upregulation as the primary mechanisms underlying increased expression of inflammatory genes, among which is RIG-1. They go on to show that treatment of EGFR-mutant lung cancer cells, and humanized xenografts with osimertinib followed by a RIG-1 agonist results in increased tumor cell death and decreased xenograft growth. Finally, in a syngeneic immunocomponent model of EGFR-mutant lung cancer, they show that the effects of RIG-1 agonist treatment are at least partially CD8+ T-cell and NK-cell dependent. Overall, this study is of high scientific interest and of potential clinical importance. However, the following points should be addressed.

We thank the reviewer for appreciating our work.

Major points:

1. The authors switch between model systems and disease models from figure to figure and this can make the manuscript difficult to follow. A model figure would be helpful to summarize their results.

We thank the reviewer for this constructive suggestion. We do incorporate a large number of different tumor models including various driving oncogenes, tumor entities, GEMM-derived cell lines, patient cohorts and mouse models. The breadth of model systems strengthens the generalizability and shows conservation of response patterns across different circumstances. However, it may at the same time make it difficult to follow if not explained well enough. We would like to apologize for a lack of clarity and added a model schematic displaying and summarizing our findings and the proposed mechanism (please see new Fig. 6g).

2. The authors initially claim that the effects of tumor cell up regulation of IFN are tumor cell autonomous, as it is not secreted into the culture media. However they also show in humanized mouse models that osimertinib treatment is associated with increased CD8+ T-cell infiltration (Fig 1f) and later show that the anti-tumor effects of a RIG-1 agonist are at least in part dependent on CD8+ T-cells and NK cells (Fig. 6e). Overall the impact of IFN-induction and RIG-1 agonist treatment on the tumor immune microenvironment remains incompletely characterized. For example, the mechanism underlying increased T-cell infiltration and how this may be affected by RIG-1 agonist treatment is not explored, nor is the affect of osimertinib or RIG-1 agonist treatment on innate immune cells within the tumor microenvironment.

This point is well taken. In order to elucidate further the potential mechanism by which immune cells are attracted into the tumor following kinase inhibition we revisited our RNA-seq data. Interestingly, we identified several cytokines and chemokines across three sets of experiments that may be involved in remodeling of the TME. Among them also several members for which chemoattractant properties have been proposed such as CXCL10, CCL5 and CXCL16.

In addition, we also observed induction of other genes known to be involved in tumor-immune-interactions such as ICAM1 (Ruscetti et al., Science 2018). Taken together, these factors may contribute to reshape the TME even in the absence of secreted IFN, but most likely possess a certain degree of heterogeneity between different models and tumor entities.

To further assess the effects on the TME evoked by osimertinib we have now performed Hyperion imaging analysis of the first cohort of humanized mice. This analysis allowed us to not only validate the higher rate of tumor infiltrating T lymphocytes, but also to show that they have an active phenotype including cytolytic activity (new Fig. 1h, Fig. S3a-c).

New Fig. 1h: Cytokeratin (CK, cyan) staining marks tumor cells, green CD8 cells and red Granzyme B expression. Overlay of green and red yields yellow signals showing concurrent staining for CD8 and Granzyme B. This indicates cytolytically active CD8 cells.

Moreover, we conducted detailed FACS analyses in a new cohort of humanized mice treated with osimertinib, a new cohort of humanized mice treated with osi/vehicle +/- IVT4 and a new cohort of syngeneic mice treated with osi +/- IVT4.

Interestingly, these analyses suggested that osimertinib reduced the expression of exhaustion markers especially on CD8 T cells, which were further decreased by IVT4 treatment. In the humanized treated with osimertinib and IVT we e.g. observed a lower proportion of TIM3+,PD1+,CD8+ cells compared to osi IVT-GAC treated tumors (new Fig. 5f). This difference was neither present in CD4 cells nor did we observe such differences in veh IVT4 vs. veh IVT-GAC tumors (new Fig. 8d). This indicates that IVT4 may contribute to a more active cytotoxic T cell response.

In the humanized mice expression of PD1 was significantly lower in CD8 cells in the osi+IVT4 arm (new Fig. 6e, MFI = mean fluorescence intensity). The number of CD8 cells per mg of tumor tissue however was unchanged and NK cells showed a slight, but non-significant increase (new Fig. S9d). Overall, this suggests that osimertinib treatment in combination with IVT4 may help to induce a less exhausted CD8 phenotype and thus lead to a stronger tumor response.

Minor points:

1. The authors cite other studies showing that MAPK signaling can repress IFN gene expression, however, the mechanism underlying this effect is unclear. Experiments to assess this should be considered.

We thank the reviewer for this suggestion. To further investigate the mechanistic basis by which MAPK signaling may contribute to IFN gene expression we have now added a series of additional experiments. First, we performed additional experiments using a specific ERK inhibitor. While inhibiting ERK led to a feedback-regulated increase of p-MEK it still increased IRF1 and IFIT1 protein levels (new Fig. S4j). This data further supports our initial hypothesis.

To investigate whether the increase of inflammatory genes like IRF1 and VTCN1 are due to de novo transcription we additionally performed ChIP-qPCR analyses following osimertinib treatment. These experiments showed that osimertinib significantly increased the occupancy of RNA Polymerase II at the transcription start site (TSS), the gene body (GB) and at the termination site (Term.) of both genes. This was true for total Pol II and also for Pol II phosphorylated at Ser2 and Ser5, which indicates transcriptional activation (new Fig. 3k, l).

This suggests that induction of IRF1 and VTCN1 is indeed driven by de novo transcription rather than e.g. altered mRNA or protein half-life. DUSP6, for which RNA expression is severely decreased following osimertinib treatment, in contrast showed strongly decreased Pol II binding indicative of transcriptional repression in alignment with decreased RNA levels (new Fig. S5i, j).

Taken together this shows that inhibition of MAPK signaling leads to relevant shift in the transcriptional output by de-emphasizing negative feedback regulators like DUSP6, while at the same time relieving the break of inflammatory genes IRF1 and VTCN1.

2. In figure 6 the authors utilize a syngeneic flank-model of EGFR-mutant lung cancer to assess the role of CD8 T-cells and NK cells in their system. An orthotopic model of lung cancer would provide a more relevant assessment of the role of the native immune-microenvironment in mediating the effects of a RIG-1 agonist.

This point is well taken. We agree that an orthotopic mouse model would be ideal to fully investigate the effects of a RIG-I agonist on the native immune-microenvironment. However, at this point the route of IVT4 administration, i.e. intratumoral injection, unfortunately currently precludes the use of an orthotopic model. Other routes of administration e.g. inhalation have been used for other immune agonists, but are not established for RIG-I agonists, for which also clinical studies have been done using intratumoral injections. The focus of our study was to investigate the adaptive response of tumor cells to targeted kinase inhibition and whether this may be exploited in combination with a RIG-I, which enabled the discovery of a novel combinatorial treatment strategy using agents that are clinically tested. Given our promising results an optimized formulation of these type of agonists or alternative chemical scaffolds warrant further investigation.

Recognizing the need for a deeper understanding of the effects on the immune microenvironment and tumor-immune interactions we performed several additional lines of experiments during the revisions including the new cohorts of humanized mice treated with osimertinib, humanized mice treated with osi/vehicle +/- IVT4 and new syngeneic mice treated with osi +/- IVT4 described in detail above. In these two independent models we derive similar conclusions with regard to the impact of IVT4 on the T cell phenotype.

Finally, we also performed a new study in a non-humanized, i.e. immunocompromised, PC9 xenograft following the set-up of the humanized model. Interestingly, we did not observe relevant effects of IVT4 (new Fig. S8f).

Together with the lack of a relevant effect in the vehicle pre-treated arms in the humanized PC9 xenografts this highlights two major points: Firstly, sustained and significant reduction of tumor volumes by IVT4 are considerably enhanced after pre-treatment with osimertinib. Secondly, this response is dependent on functional immune components, which is in line with the reduced effects of osi+IVT4 following CD8 and NK cell depletion in the syngeneic mice.

Taken together, our data suggest that the combination of targeted kinase inhibition enables an inflammatory reprogramming of both tumor cells and its micro-environment, which in a one-two punch approach can be exploited using a RIG-I agonist. It thereby establishes a novel treatment strategy that warrants further investigation across different tumor entities.

Reviewers' Comments:

Reviewer #1:

Remarks to the Author:

The authors have addressed all of my concerns. I recommend publication.

Reviewer #2:

Remarks to the Author:

The manuscript entitled "MAPK-mediated inflammatory reprogramming sensitizes tumors to targeted

activation of innate immunity sensor RIG-I" by Brägelmann et al. studies the causes of treatment-induced senescence, in an effort to undermine the mechanisms by which cancers escape treatment.

Beginning with transcriptomic analyses, the authors studied oncogene-driven cancer cells following treatment with driver-specific kinase inhibitors. The authors report a converse relationship between

proliferative gene signatures (e.g., cMyc, E2F1, and G2/M signatures) and innate immune gene signatures (IFN α , IFN γ and adult tissue stem cell (ATSC) gene signatures, along with a subset of known senescence-associated genes. The authors suggest that signaling through MAPK is sufficient to

push this inverse transcriptomic balance towards increased proliferation with decreased innate immunity, while MAPK inhibition (using either a MAPK inhibitor or an inhibitor that blocks an upstream

activator of MAPK, like EGFR or HER2 or MET) is sufficient to tip the balance towards greater innate

immunity and less proliferation. The authors suggest that inhibition of proliferation or induction of cell

death is not required for innate immune activation, however. Further, the authors show that agonist-inducible RIG-I signaling can be used to amplify the innate immune activation in conjunction with

EGFR/MAPK inhibition to improve tumor growth inhibition.

The manuscript represents a good deal of data that is of high quality. The major attributes of the manuscript are the strong data showing combined efficacy of EGFR inhibition + RIG-I activation in lung

tumors, which is a timely advancement for a very pressing and unmet need. The major criticisms from the original

manuscript version, primarily over-interpretation of the data in many places, have been addressed in the revised version. The Reviewer is satisfied with the revised version of the manuscript, and consider this body of work a timely and impactful study aiming to improve the outcome of cancer patients, while expanding our understanding of the complexities of tumor immunology.

Reviewer #3:

Remarks to the Author:

The authors have thoroughly addressed my concerns.

REVIEWERS' COMMENTS

Reviewer #1 (Remarks to the Author):

The authors have addressed all of my concerns. I recommend publication.

We thank the reviewer for the appreciation of our work and for the constructive comments in the first round of review.

Reviewer #2 (Remarks to the Author):

The manuscript entitled "MAPK-mediated inflammatory reprogramming sensitizes tumors to targeted activation of innate immunity sensor RIG-I" by Brägelmann et al. studies the causes of treatment-induced senescence, in an effort to undermine the mechanisms by which cancers escape treatment. Beginning with transcriptomic analyses, the authors studied oncogene-driven cancer cells following treatment with driver-specific kinase inhibitors. The authors report a converse relationship between proliferative gene signatures (e.g., cMyc, E2F1, and G2/M signatures) and innate immune gene signatures (IFN α , IFN γ and adult tissue stem cell (ATSC) gene signatures, along with a subset of known senescence-associated genes. The authors suggest that signaling through MAPK is sufficient to push this inverse transcriptomic balance towards increased proliferation with decreased innate immunity, while MAPK inhibition (using either a MAPK inhibitor or an inhibitor that blocks an upstream activator of MAPK, like EGFR or HER2 or MET) is sufficient to tip the balance towards greater innate immunity and less proliferation. The authors suggest that inhibition of proliferation or induction of cell death is not required for innate immune activation, however. Further, the authors show that agonist-inducible RIG-I signaling can be used to amplify the innate immune activation in conjunction with EGFR/MAPK inhibition to improve tumor growth inhibition.

The manuscript represents a good deal of data that is of high quality. The major attributes of the manuscript are the strong data showing combined efficacy of EGFR inhibition + RIG-I activation in lung tumors, which is a timely advancement for a very pressing and unmet need. The major criticisms from the original

manuscript version, primarily over-interpretation of the data in many places, have been addressed in the revised version. The Reviewer is satisfied with the revised version of the manuscript, and consider this body of work a timely and impactful study aiming to improve the outcome of cancer patients, while expanding our understanding of the the complexities of tumor immununology.

We thank the reviewer for the appreciation of our work and for the very insightful comments and experimental suggestions during the review process that helped to strengthen the conclusions drawn.

Reviewer #3 (Remarks to the Author):

The authors have thoroughly addressed my concerns.

We thank the reviewer for the appreciation of our work.